# Embryogenic callus induction from immature zygotic embryos and genetic transformation of *Larix kaempferi* 3x *Larix gmelinii* 9

**Sufang Zhang[1], Shanshan Yan[1], Peiqi An[1], Qing Cao[1], Chen Wang[1], Junhui Wang[2], Hanguo Zhang[1]\*, Lei Zhang[1]\***

1 State Key Laboratory of Tree Genetics and Breeding (Northeast Forestry University), Harbin, China,
2 State Key Laboratory of Tree Genetics and Breeding (Chinese Academy of Forestry), Beijing, China

\* hanguozhang1@sina.com (HZ); zhanglei@nefu.edu.cn (LZ)

**Data Availability Statement:** All data sustaining the results in this study are included in this manuscript or its Supplementary information files.

## Abstract

To date, there are few reports of the successful genetic transformation of larch and other conifers, mainly because it is difficult to transform and integrate exogenous genes. In this study, hybrid larch *Larix kaempferi* 3x *Larix gmelinii* 9 cones were collected on June 27, July 1, July 4, July 7 and July 16, 2017. Embryogenic callus induction was studied using a combination of different plant growth regulators and concentrations. The results showed that July 1 was the best stage; the highest induction rate was 10.83%, which cultured in BM medium (Button medium, which formula was listed in S1 Table) with 1.0 mg/L 2,4-D (2,4-dichlorophenoxyacetic acid) and 0.2 mg/L KT(kinetin). When cultured on a proliferation medium for 12 days, proliferation was the fastest, reaching 323.08%, which could also maintain the freshness and vitality. The suitable pre-culture medium for somatic embryogenesis was 1/4 BM medium containing 10 g/L inositol and 60 g/L sucrose. The combination of 45 mg/L ABA (abscisic acid) and 75 g/L $PEG_{4000}$ (Polyethyene glycol 4000) could promote the number of somatic embryos, and reached the maximum, 210 140 per 1 g FW. The genetic transformation was carried out by the *Agrobacterium*-mediated transformation method with embryogenic callus cultured for 12 days. The results showed the optimal $OD_{600}$ of the infection solution(suspension of *A. tumefaciens*) was 0.5, co-culture time was 2 days, and screening concentration of Hyg (hygromycin B) was 4 mg/L. In this study, the transformation rate of resistance callus was 32.1%. It provides a reference for low genetic transformation efficiency of larch at present. This study could be beneficial for the innovation and breeding of larch by genetic engineering and provides a certain basis for rapid propagation of excellent larch germplasm resources and genetic engineering breeding of larch and other conifers.

## Introduction

Larch is among the main genus in the representative cold-humid coniferous forest. Because of its wide distribution, strong adaptability, fewer pests and diseases, straight trunks and excellent materials, larch has become among the main afforestation species in Northeast China [1]; due

**Funding:** This manuscript was supported by the National Science and Technology Major Project (2018ZX08020003-001-001); the National Natural Science Foundation of China (31700595); the Fundamental Research Funds for the Central Universities (2572019BA13).

**Competing interests:** The authors have declared that no competing interests exist.

**Abbreviations:** 2,4-D, 2,4-dichlorophenoxyacetic acid; ABA, abscisic acid; As, acetosyringone; BA, 6-Benzylaminopurine; Cef, cefotaxime; CH, acid hydrolysed casein; EC, embryogenic callus; ESM, embryogenic suspensor mass; FW, Fresh Weight; Gen, gentamicin; Gln, glutamine; Hyg, hygromycin B; Kan, kanamycin; KT, kinetin; MG, megagametophyte; NAA, 1-naphthaleneacetic acid; NEC, non-embryogenic callus; PEG$_{4000}$, Polyethyene glycol 4000; PEM, pro-embryogenic mass; SE, Somatic embryogenesis.

to its early fast-growth, larch has emerged from all conifers and is an important pulp and wood species in Northeast China, North China and the subalpine region of South China. With a globally forest cover decreasing, there has been an increased focus on developing genetically improved strains of Larix in order to accelerate afforestation in recent years. Although many excellent larch varieties have been selected by conventional breeding methods, its long growth cycle, the high mutation rate of offspring during sexual reproduction, and low asexual reproduction rate of cuttings make it very difficult to multiply and apply in the short term. Therefore, conventional breeding methods can not achieve rapid genetic improvement of Larch. Currently, somatic embryogenesis has a strong reproductive capacity, complete structure and not affected by external factors, such as seasons, which undoubtedly provides more effective solutions to the above difficulties. Regarding *Larix* spec., several studies have been conducted over the last few decades [2]. Since embryos of *L.x eurolepis* and *L.x leptoeuropaea* at precotyledonary developmental stage were used as an explant to induce embryogenic callus, somatic embryogenesis of larch began to develop rapidly [3]. *Larix decidua* [4], *Larix occidentalis* [5], *Larix kaempferi* [6], *Larix principis-rupprechtii* [7], *Larix* x *eurolepis* and *Larix* x *marschlinsii* [8] and *Larix olgensis* [9] all established a more stable somatic embryogenesis system. Previous studies have found that the induction of embryogenic callus is related to many factors and that the types and concentrations of plant hormones have a strong influence on the induction of embryogenic callus. In addition, auxin 2,4-D and NAA (1-naphthaleneacetic acid), cytokinins BA (6-Benzylaminopurine) and KT [10], which is commonly used in conifers. Embryogenic callus induction medium with an appropriate concentration of auxin can increase the induction rate [11]. Furthermore, studies have shown that explants at different developmental stages have different induction rates. Lelu found that in immature zygotic embryos of hybrid larch used as explants, the induction rate of embryogenic callus was significantly higher than that of mature zygotic embryos [8]. Regarding somatic embryogenesis studies including western larch, although 21~93% of the experiments can induce callus, only 3% of the experiments can continue to induce embryogenic cell lines and form mature somatic embryos, which requires obtaining a more stable embryogenic callus and stable proliferation to achieve better subsequent test results. Reports regarding the somatic embryogenesis of hybrid larch, which possesses economical relevant heterosis effects at home and abroad, are rare. Previous studies showed hybrid larch (*Larix* x *eurolepis* Henry) frequently exhibits a superior growth over inclusive environments, when compared to pure parental species [12]. Although immature zygotic embryos have been used as explants in hybrid larch, the induction rate of embryogenic callus was low, the induction rate of callus weaving was up to 100% under suitable conditions in hybrid larch, but the highest induction rate of embryogenic callus was only 9.2% [13]. As is well known, the premise of plant somatic embryogenesis is that embryogenic callus is induced and can proliferate stably; thus, it is necessary to induce stably proliferating embryogenic callus in hybrid larch.

On the other hand, somatic embryogenesis is currently used for clonal propagation, mainly utilized in forestry and forestry research, and it is also an enabling tool for genetic transformation. Transgenesis is a promising and effective approach for accelerating the genetic improvement of forest trees [14]. The regenerative ability of the recipient materials is a major determinant of genetic transformation [15]. Many studies showed that the embryogenic callus is an ideal material for transgenesis on account of its stable proliferation, high regeneration rate, and tolerance to *Agrobacterium tumefacien* [16]. As early as 1934, *Agrobacterium tumefaciens* was discovered can infect conifers, and studies have shown that coniferylol extracted from pine can promote conifers infection [17]. In the late 1980s, 16 seedlings of conifers were used as materials to study the ability of different strains of *Agrobacterium tumefaciens* to inoculate crown tumours. The results showed that although different strains had large differences in

their tumourigenic ability, the conifers are indeed the natural host of *Agrobacterium*, which laid a theoretical foundation for the *Agrobacterium*-mediated genetic transformation in conifers [18,19]. Exogenous genes were transformed to *Pinus taeda* by *Agrobacterium*-mediated method, which was the first report on the conifers transformed successfully [20]. Since 1991, the first genetically transformed conifer, European larch, was obtained, and the genetic transformation of larch had been developed [21]. The hypocotyls of European larch zygotic embryos as explant by *Agrobacterium*-mediated method and was the first to transfer the commercially valuable B.t gene into European larch to obtain transgenic plants [22]. The embryogenic callus of *L.* x *eurolepis* and *L.* x *leptoeuropea* were transformed by *Agrobacterium tumefacien* and obtained resistance callus, but the transformation efficiency was extremely low, only 1%~2% [23].

However, regarding current research on the genetic transformation of larch, especially hybrids, is still in its infancy and is far from meeting the requirements of genetic improvement technology and commercial production. Therefore, in this study, immature zygotic embryos of hybrid larch *Larix kaempferi* 3 x *Larix gmelinii* 9 were used for embryogenic callus induction, thus can provide a rapid propagation of excellent larch germplasm resources, and then embryogenic callus as materials were uesd to transform through *Agrobacterium tumefacien*, which can not only accelerate the development of superior transgenic strains of hybrid larch, but also provid a basis for gene functional studies and a reference for genetic engineering breeding of other larch and conifers.

## Results

### Optimal sterilisation conditions for explants

Take 3% NaClO surface sterilisation at different time as the experiment, and the contamination rate and survival rate were calculated after three weeks. The results showed that (Table 1) the contamination rate and survival rate ($p = 0.000$) decrease significantly with the extension of the sterilisation time. Disinfected for 8 min, the contamination rate was high. When the sterilisation proceeded for 18 min, although the contamination rate was not high, most seeds were killed due to the long sterilisation time. When the sterilisation proceeded for 20 minutes, the phenomenon was more obvious, largely because the sterilisation time was too long, the seed was killed, and the survival rate was only 2%. Compared with sterilisation for 8 min, the contamination rate of sterilisation for 10~20 min was lower, and the difference was significant. However, the contamination rate was very low at 10 min, 12 min and 15 min, and the survival rate was greatly improved. Although the contamination rate was lower at 12 min and 15 min, considering the final survival rate of the seeds, the seeds were the least damaged when disinfected for 10 min. And the difference between disinfection 10 min and other sterilisation time is significant. Therefore, the best sterilisation method was as follows: first, after peeling the seed coat, the explants were placed in a 4˚C refrigerator for 12 hours and then washed with sterile deionized water 3 times; second, the samples were disinfected with 75% alcohol for 1 min and washed with sterile deionized water 3 times; finally, the samples were disinfected with 3% NaClO for 10 min and then washed with sterile deionized water 3 times.

### Induction of embryogenic callus

**Embryogenic callus status.** The zygotic embryos (Fig 1A and 1B) collected from June 27 to July 4, 2017, were mostly in precotyledonary developmental stage,(about 58~65 days after pollination)and the embryo and endosperm were not easily separated, which was based on the definition of development stages [24]. The inoculation was carried out as follows:(1) the embryo and endosperm were longitudinally cut together with a downward incision, and then,

**Table 1. Effects of the 3% NaClO sterilisation treatment on the contamination and survival rates.**

| Sterilisation time (min) | Inoculation number | Contamination rate (%) | Survival rate (%) |
|---|---|---|---|
| 8 | 200 | 30±4 a | 40±9 b |
| 10 | 200 | 4±1 b | 90±15 a |
| 12 | 200 | 3±1 bc | 80±15a |
| 15 | 200 | 2±1 bc | 65±22 ab |
| 18 | 200 | 2±2 bc | 40±11 b |
| 20 | 200 | 0 c | 2±2 c |

Notes: The values are mean±SD, and different letters indicate significant differences at 0.05 level.

they were inoculated in embryogenic callus induction medium. Most embryos collected from July 7 to July 16 were relatively mature(68~77 days after pollination), and the embryo and endosperm were easily separated. Inoculation was carried out as follows:(2) only the embryos were selected and inoculated on the medium. One month after inoculation, only a few explants formed white transparent or yellowish callus on the surface after the incision, and the yellowish callus was browned and died in the subsequent culture; the white, transparent callus became harder, brown, and eventually necrotic or continued to proliferate. These white with hard or yellowish calli tissues were observed under a microscope and were found to have no embryogenic suspensor cell clusters that were non-embryonic callus (Fig 1C, NEC) and non-embryonic callus structure (Fig 1D). Non-embryogenic callus of friable consistence was composed of round-shaped cells with prominent nuclei. The callus was prone to browning and necrosis, and cultivation of this callus on maturation medium resulted in tissue necrosis. After

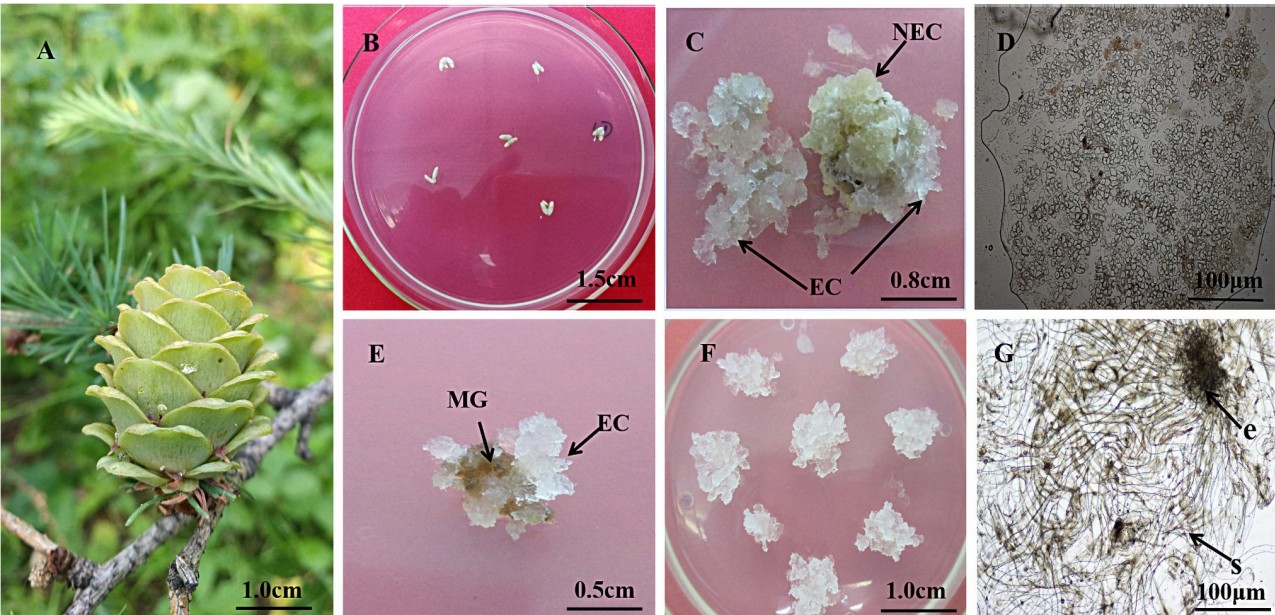

**Fig 1. Induction and microscopic structure of embryogenic callus.** A: Cones of immature larch; B: Inoculation of explants; C: Callus(EC: Embryogenic callus; Non-embryogenic callus, NEC), D: Microstructure of non-embryogenic callus; E: Initiation of EC-white mucilaginous tissue protruded from the micropylar end of the megagametophyte (MG) explant on culture medium BM containing 1.0 mg/L 2,4-D and 0.2 mg/L KT; F: Embryogenic tissue cultured on proliferation medium; G: Microstructure of embryogenic callus (ESM, embryogenic suspensor mass- e: Embryogenic cells; s: Suspensor similar cells).

4~6 weeks culture, the explants on the initiation medium gave rise to embryogenic tissue extruding from the micropylar end of the megagametophyte (Fig 1E, MG and EC). In addition, the separation of embryogenic callus from primary explants and further culturing on the same medium with plant growth regulator resulted in stable proliferation (Fig 1F). The surface of EC formed a colourless and transparent filiform protuberance, which grew very slowly and was found to have an embryogenic suspensor cell cluster structure (Fig 1G, e and s) [25]. EC had the potential to form somatic embryogenic tissue(Fig 1C, EC, 1E and EC); however, it was easily affected by non-embryonic callus(Fig 1C, NEC). Embryonic callus that grew together with non-embryonic callus may be transformed into the non-embryogenic callus. Thus, if embryogenic callus was not quickly picked out from non-embryonic callus, it could soon be affected by non-embryonic callus and lose embryogenicity.

**Effect of plant growth regulators on embryogenic callus induction.** The larch cones collected on July 1 were selected, and the seed scales and seed coats were peeled; then, the zygotic embryos were disinfected and placed in induction medium, and the induction rate was calculated after one month (Table 2). The results showed that if the 2,4-D concentration was too low (0.2 mg/L) or too high (2.5 mg/L), then embryogenic callus could not be induced, and the induction rate increased as the 2,4-D concentration increased. The induction rate reached maximum when the 2,4-D concentration reached 1.0 mg/L. However, the induction rate was reduced when the concentration was more than 1.0 mg/L. The results of different plant growth regulator combinations showed that 1.0 mg/L 2,4-D and 0.2 mg/L KT added in the medium had the highest induction rate, and the difference was significant when the concentration of 6-BA and KT was greater than or equal to 0.5 mg/L, indicating that lower concentration of 6-BA and KT may be required for induction of hybrid larch embryogenic callus. Part of the appropriate concentration of plant growth regulators can also induce embryogenic callus, but the induction rate is not high. Through the analysis of variance of the induction rate under each combination, the results showed that each combination has a significant influence on the induction rate ($p = 0.000$). Therefore, the optimal combination of hormones for the induction of embryogenic callus in hybrid larch *Larix kaempferi* 3x *Larix gmelinii* 9 was 1.0 mg/L2, 4-D and 0.2 mg/L KT.

## Proliferation of embryogenic callus

2,4-D is important for the expression of larch embryonic competence. Following long-term subculture in medium containing high concentration 2,4-D, embryogenic callus can form a large number of early embryos, but the development of the embryo will be inhibited, resulting in the loss of somatic embryo maturation ability and a decrease in the quality of the somatic embryos. However, without auxin, embryogenic callus will die, in this study, concentration of 2,4-D for proliferation was its 1/2, 1/5 and 1/10 times on the induction medium.

The induced embryogenic callus can be divided into two types according to morphology (Fig 2): the first type is loose and fragile, crystal clear, obvious granular, and has a strong division ability (Fig 2A), which is the ideal receptor material for transgene. The second type is more compact, milky white translucent, watery, has a certain division ability, but the ability is weaker than the first type (Fig 2B). After the callus was induced, it entered the three stages of the original embryo because of its continuous division and proliferation (Fig 2C–2E). The pro-embryogenic mass(PEM)stage I PEMI (Fig 2C) contained two embryonic suspensor structures, one embryogenic cell, the pro-embryogenic mass stage II PEMII (Fig 2D) contained three embryonic suspensor structures, the embryogenic cell was enlarged, and pro-embryogenic mass stage III PEMIII (Fig 2E) contained many embryonic suspensor structures and the structures elongated compared with PEMII, and the embryogenic cell continued to

**Table 2. Effect of plant growth regulators on embryogenic callus induction.**

| Medium number | Inoculation number | 2,4-D concentration (mg/L) | BA concentration (mg/L) | KT concentration (mg/L) | Embryogenic callus cell lines number | Induction rate (%) |
|---|---|---|---|---|---|---|
| 1 | 120 | 0.2 | 0 | 0 | 0 | 0 e |
| 2 | 120 | 0.2 | 0.2 | 0.2 | 0 | 0 e |
| 3 | 120 | 0.2 | 0.5 | 0.5 | 0 | 0 e |
| 4 | 120 | 0.2 | 1 | 1 | 0 | 0 e |
| 5 | 120 | 1 | 0 | 0.2 | 13 | 10.83±2.5 a |
| 6 | 120 | 1 | 0.2 | 0 | 8 | 6.67±3.34 b |
| 7 | 120 | 1 | 0.5 | 1 | 3 | 2.5±1.67 de |
| 8 | 120 | 1 | 1 | 0.5 | 2 | 1.67±1.67 de |
| 9 | 120 | 1.8 | 0 | 0.5 | 7 | 5.83±1.67 bc |
| 10 | 120 | 1.8 | 0.2 | 1 | 4 | 3.33±2.5 cd |
| 11 | 120 | 1.8 | 0.5 | 0 | 2 | 1.67±1.67 de |
| 12 | 120 | 1.8 | 1 | 0.2 | 5 | 4.17±1.67 bcd |
| 13 | 120 | 2.5 | 0 | 1 | 0 | 0 e |
| 14 | 120 | 2.5 | 0.2 | 0.5 | 0 | 0 e |
| 15 | 120 | 2.5 | 0.5 | 0.2 | 0 | 0 e |
| 16 | 120 | 2.5 | 1 | 0 | 0 | 0 e |

Notes: The values are mean±SD, and different letters indicate significant differences at 0.05 level.

proliferation. In this study, it was found that when the callus was the first type, many more calli would enter the PEMIII, and the second type would gradually die with the decrease of the division ability during the subculture. Therefore, the first type callus was more suitable for follow-up research.

However, the newly induced embryogenic callus is extremely unstable and will have different development states in the first several subcultures. Among them, most callus will lose embryogenicity and then transform into non-embryonic callus [26]. Therefore, it is necessary to screen the conditions necessary for the stable proliferation of embryogenic callus and determine the optimal subculture cycle.

The vigorously growing callus was inoculated on proliferation medium; the fresh weight was weighed every 3 days, the growth state of the callus was observed until the callus entered a recession period, and each treatment was repeated 3 times. After an adaptation period of 0~9 days, the callus quickly entered the logarithmic growth period. However, the optimal subculture time was relatively consistent, all within 12~18 days. The fresh weight of the callus reached the maximum, i.e., 1373.33 mg (the initial inoculation amount, ie, at day 0 was 130 mg/cluster embryogenic callus, so the proliferation rate was 956.41%) on proliferation medium 1/2-BM (the hormone combination was 2,4-D 0.5 mg/L+KT 0.2 mg/L) for 33 days. Subsequently, proliferation began to show a negative growth trend. Moreover, proliferation was the fastest when cultured on this medium for 12 days (87.78 mg/day), and the proliferation rate at this time was also high compared with the fresh mass weight at 0 day, reaching 323.08%, which can not only maintain freshness and vitality but also leads to a large amount of proliferation as the best subculture time (Fig 3).

### Somatic embryo maturation and germination

**Effect of pre-culture on somatic embryogenesis.** This study found that embryogenic calli almost no longer proliferate on the transition medium without plant growth regulators,

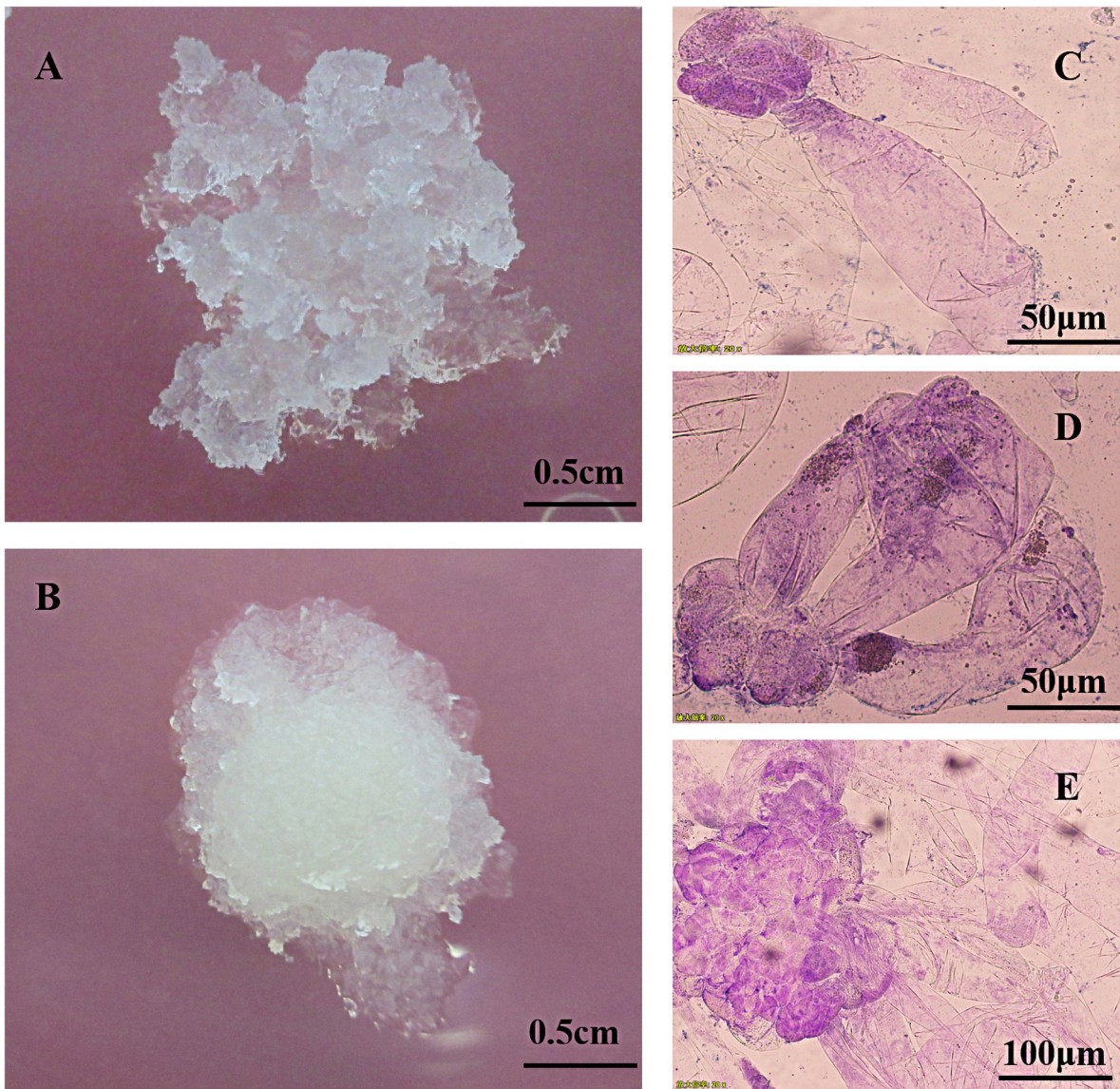

**Fig 2. Proliferating embryogenic callus and protoembryo microscopy structure.** A, B: Proliferating embryogenic callus; C: Pro-embryogenic mass I (PEMI); D: Pro-embryogenic mass II (PEMII); E: Pro-embryogenic mass III (PEMIII).

but form protruding filamentous structures on the surface of the callus. When the ion concentration of the medium decreases significantly, it is conducive to somatic embryogenesis, and increases with the decrease of ion concentration. In general, inositol has a greater impact on somatic embryos. As the concentration increases of inositol, the number of somatic embryos increases first and then decreases. When the concentration of inositol increased by 10 times, the number of somatic embryos in all media reached the maximum, and when the medium was 1/4 BM and the concentration of sucrose was 60 g/L, the number of somatic embryos was the highest, which was 140 per 1 g. Through the analysis of variance on the number of somatic embryos cultured on the same medium, it was found that sucrose and inositol had a significant effect when the medium was 1/4 BM ($p = 0.042 < 0.05$), but little effect under the other two mediums($p = 0.125$, $p = 0.096$), multiple comparison analysis found that when the medium

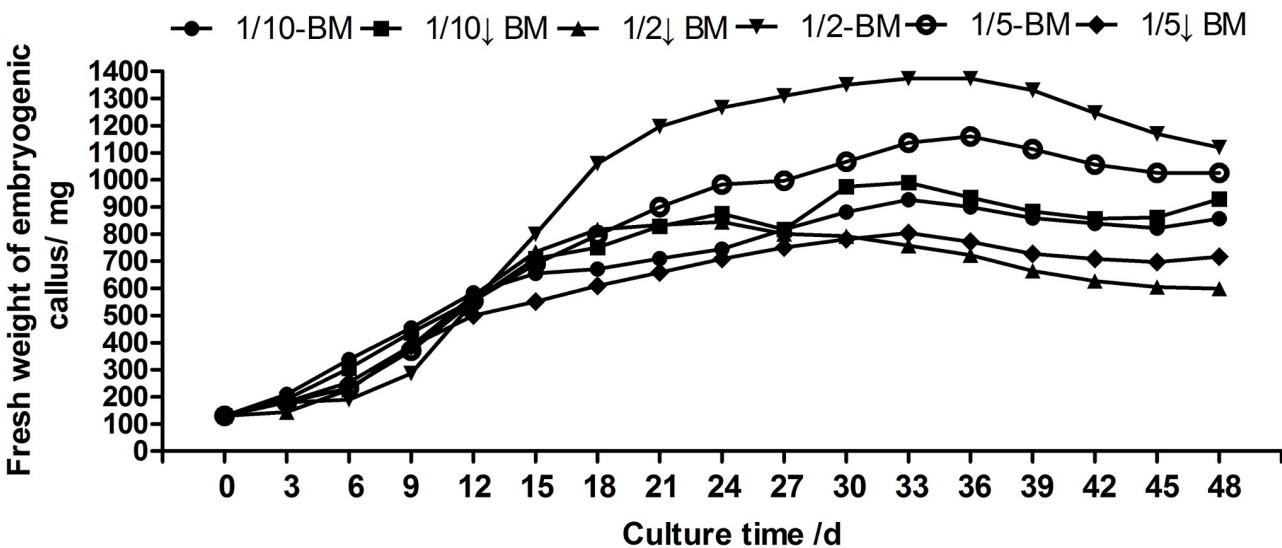

**Fig 3. Growth curve of embryogenic callus on different proliferation media.**

was 1/4 BM, the sucrose concentration was greater than or equal to 60 g/L, and the inositol concentration was greater than or equal to 10 g/L, the combination has better results than other combinations. The number of somatic embryos was higher and the difference was significant (Table 3). In general, the best pre-culture medium for somatic embryogenesis was 1/4 BM medium containing 10 g/L inositol and 60 g/L sucrose.

**Effect of ABA and PEG$_{4000}$ on somatic embryogenesis..** After the somatic embryo was induced and cultured for about one week, the surface of the callus gradually became dry, and the filamentous structure began to thicken (Fig 4A). After about two weeks of culture, a proembryo with spherical shape can be seen under the microscope (Fig 4B). Continue culture, the embryo stalk of the spherical shape proembryo gradually faded away, and the embryo head gradually became larger and transformed into a yellow structure (Fig 4C). When cultured for about four weeks, a clear cotyledon embryo structure was formed, and the existing callus began to brown and died. About 6~8 weeks, the cotyledon embryo reached maturity (Fig 4D),

**Table 3. Effects of medium ion concentration, sucrose and inositol on the number of somatic embryos.**

| Somatic embryogenesis medium number | Initial inoculation amount (mg) | BM multiple (ion concentration) | Sucrose(g/L) | Inositol(g/L) | Number of somatic embryos/g FW |
|---|---|---|---|---|---|
| 1 | 200 | 1/4 (25%) | 30 | 1 | 75±15 bc |
| 2 | 200 | 1/4 (25%) | 60 | 10 | 140±35 a |
| 3 | 200 | 1/4 (25%) | 90 | 15 | 125±20 a |
| 4 | 200 | 1/2 (50%) | 30 | 10 | 85±30 b |
| 5 | 200 | 1/2 (50%) | 60 | 15 | 70±10 bc |
| 6 | 200 | 1/2 (50%) | 90 | 1 | 45±15 cd |
| 7 | 200 | 1 (100%) | 30 | 15 | 45±5 cd |
| 8 | 200 | 1 (100%) | 60 | 1 | 30±15 d |
| 9 | 200 | 1 (100%) | 90 | 10 | 50±5 bcd |

Notes: The values are mean±SD, and different letters indicate significant differences at 0.05 level; FW: Fresh Weight; The ion concentration of water Deionized was 0%; Ions contained in BM medium was 100%.

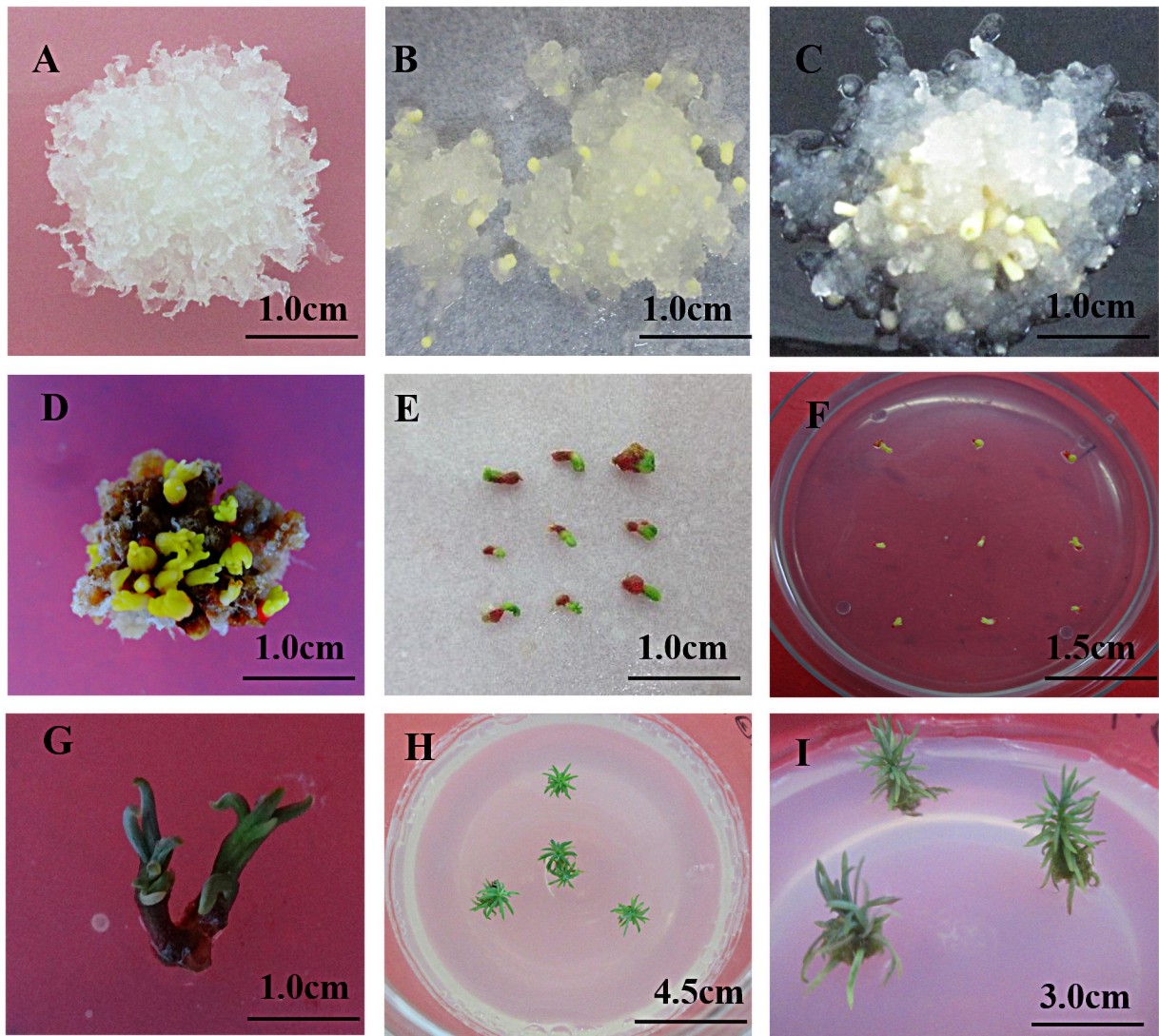

**Fig 4. Somatic embryogenesis.** A: Pre-culture; B~D: Somatic embryo induction; E: Desiccation; F~I: Germination and regeneration.

it will be "dried" by "drying" treatment(somatic embryos were placed in a sterile culture dish covered with sterile filter paper and dried for 10 minutes) to make it reach the true "physiological" mature state from morphological maturity. Then, put it on the germination medium for light germination culture. It was found that the cotyledons began to turn from yellow to green about a week, and the radicles turned red (Fig 4E and 4F). After 2 to 3 weeks, the cotyledons and radicles continued to grow in two directions respectively (Fig 4G). The cotyledon grew elongated, and the root tip growed and elongated from the radicle to form regenerated plantlets (Fig 4H and 4I). The microscopic structure of the entire maturation and germination process were shown in the Fig 5 (Fig 5, detailed annotation information was included in the legend).

The number of somatic embryos induced when different concentrations of ABA and $PEG_{4000}$ were added to the somatic embryo induction medium were counted (Table 4), and it was found that the two combination had a greater impact on somatic embryo induction. The

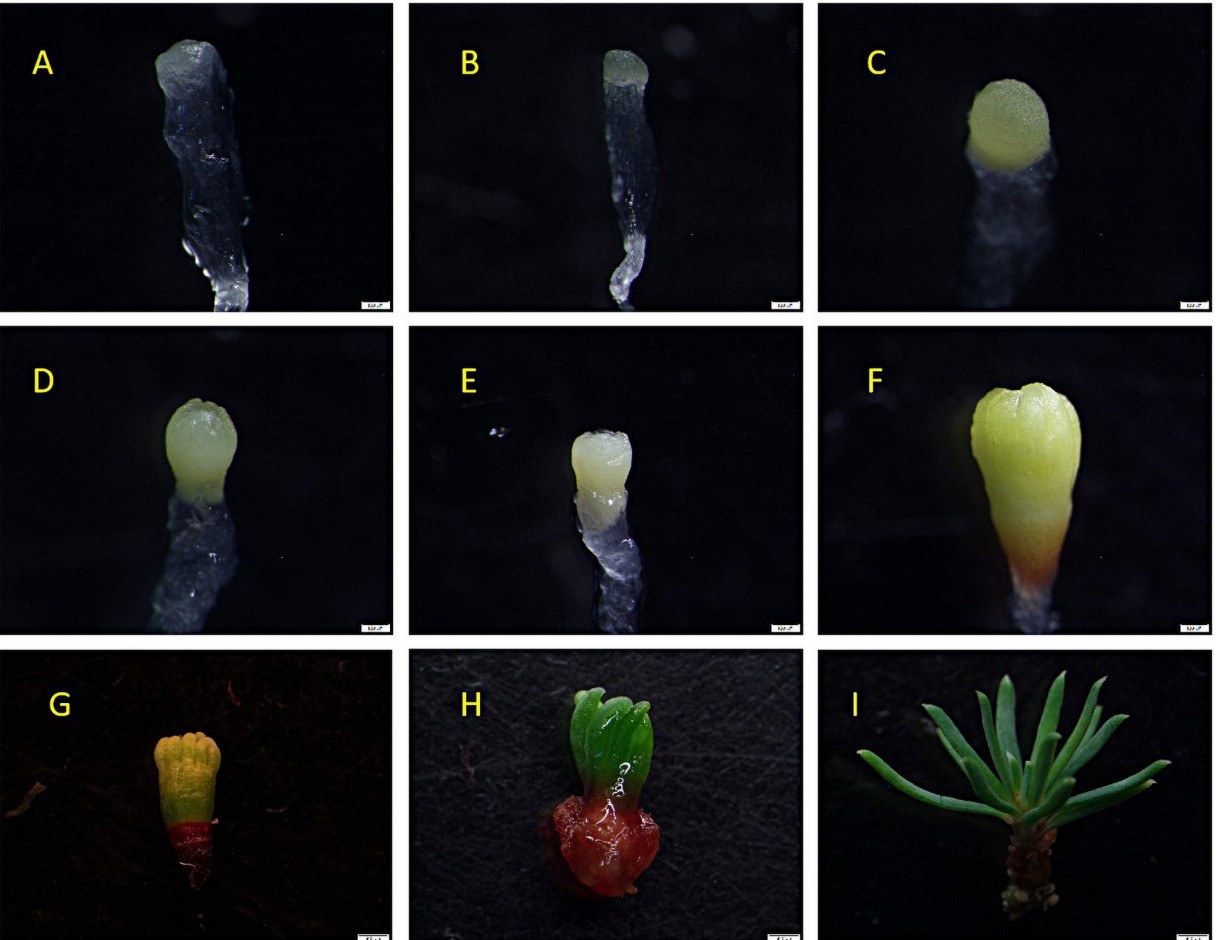

**Fig 5. Microscopic examination of larch somatic embryo maturation and germination.** A~F: The somatic embryos were cultured maturation on maturation medium 1 w, 2 w, 3 w, 4 w, 5 w, 6 w, respectively. G, H: The states of drying treatment 1 w and 2 w, respectively. I: The state of germination 4 w.

effect of $PEG_{4000}$ on the number of somatic embryos under the same ABA treatment was analyzed, and it was found that the effect of $PEG_{4000}$ on the number of somatic embryos when the ABA concentration was 15 mg/L, 30 mg/L, 45 mg/L, and 60 mg/L, respectively. All were extremely significant ($p = 0.005$, $p = 0.005$, $p = 0.000$, $p = 0.003$). Under the same $PEG_{4000}$ treatment, the effect of ABA on the number of somatic embryos was analyzed by variance analysis, and it was found that the effects of different concentrations of ABA on the number of somatic embryos at $PEG_{4000}$ concentrations of 0 g/L and 75 g/L were extremely significant ($p = 0.003$, $p = 0.001$), and the effects of different concentrations of ABA at 50 g/L and 100 g/L on the number of somatic embryos were not significantly different ($p = 0.029$, $p = 0.147$). Overall, the number of developed somatic embryos had an increasing trend with the increase of ABA concentration. When the ABA concentration reached 30~45 mg/L, the number of developed somatic embryos was more, and when ABA was 45 mg/L, it reached maximum. Under the same ABA concentration, with the increase of $PEG_{4000}$ concentration, the number of developed somatic embryos first increased and then decreased. When $PEG4_{000}$ was 50~75 g/L, the number of somatic embryos produced was larger, and when the ABA concentration was less than 45 mg and at the same ABA concentration,, the number of somatic embryos was the

**Table 4. Effects of ABA and PEG$_{4000}$ on the number of somatic embryos.**

| Medium number | Inoculation amount(mg) | ABA (mg/L) | PEG$_{4000}$ (g/L) | Number of somatic embryos/g FW |
|---|---|---|---|---|
| 1 | 200 | 15 | 0 | 20±10 f |
| 2 | 200 | 15 | 50 | 65±20 de |
| 3 | 200 | 15 | 75 | 85±25 cd |
| 4 | 200 | 15 | 100 | 30±5 ef |
| 5 | 200 | 30 | 0 | 55±20 def |
| 6 | 200 | 30 | 50 | 110±25 bc |
| 7 | 200 | 30 | 75 | 125±30 b |
| 8 | 200 | 30 | 100 | 45±10 def |
| 9 | 200 | 45 | 0 | 80±15 cd |
| 10 | 200 | 45 | 50 | 145±25 b |
| 11 | 200 | 45 | 75 | 210±25 a |
| 12 | 200 | 45 | 100 | 55±20 def |
| 13 | 200 | 60 | 0 | 15±15 f |
| 14 | 200 | 60 | 50 | 115±30 bcf |
| 15 | 200 | 60 | 75 | 80±30 cd |
| 16 | 200 | 60 | 100 | 35±10 e |

Notes: The values are mean±SD, and different letters indicate significant differences at 0.05 level.

highest when PEG$_{4000}$ was 75 g/L, and when PEG$_{4000}$ was 0, the number of somatic embryos was lower. The combination of 45 mg/L ABA and 75 g/L PEG$_{4000}$ could promote the number of somatic embryos and reached the maximum, which is 210 per 1 g fresh weight. Multiple comparisons showed significant differences that the number of somatic embryos in this combination was compared with other combinations. 50 cotyledon embryos were selected for germinating, after drying treatment, 42 somatic embryos germinated, and the germination rate was 84%.

## Optimal genetic transformation system

**Sensitivity to antibiotics.** The effect of hyg on the proliferation of embryogenic callus in larch was obvious (Fig 6A). Within 21 days, embryogenic callus growing on the proliferation medium without hygromycin continued to grow, but when the proliferation medium contained 2 mg/L hyg, there was a slight decrease during the early stage of the culture and growth increased over time, indicating that larch was sensitive to hyg. On proliferation medium

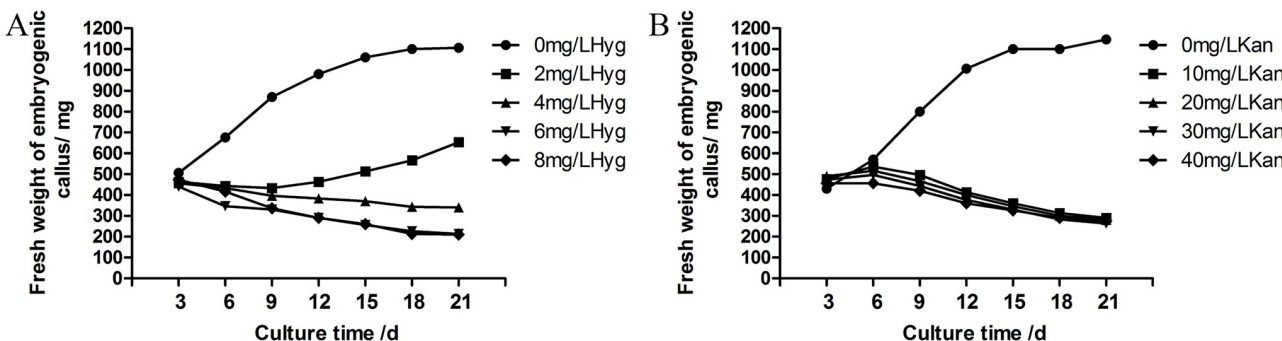

**Fig 6. Embryonic callus growth curve in proliferation medium containing different concentrations of hyg (A) and kan (B) over 21 days.**

containing 4 mg/L hyg, the growth of embryogenic callus was inhibited and continued to decrease during this period, especially in the presence of 6 mg/L and 8 mg/L hyg. Therefore, 4 mg/L hyg was a suitable screening concentration for larch genetic transformation.

Similar to hyg, the effect of kan on the proliferation of embryogenic callus in larch was also obvious (Fig 6B). Within 21 days, embryogenic callus growing on the proliferation medium without kan continued to grow, but on proliferation medium containing 10 mg/L kan, there was a slight decrease during this period. When the kan concentration was higher than 20 mg/ L, the growth of embryogenic callus was inhibited; the callus on the side touching the medium quickly browned and died, especially at 30 mg/L and 40 mg/L kan. Therefore, 20 mg/L kan was a suitable screening concentration for larch genetic transformation. Compared to hyg, the effect of kan may be less obvious, and the lower concentration of hyg contained in the medium could inhibit the growth of the callus. In addition, there have been studies showing that hygromycin was relatively toxic to plants and can inhibit the growth of plant tissues in a relatively short or short period of time, and its screening effect was relatively good. Hygromycin was used to screen transformed tissues, and showing all PCR tests were positive [27] and kanamycin screening was only 10% positive [28,29]. Therefore, in this study, an empty vector pCAMBIA1301 containing the hyg marker gene *hpt* was used for genetic transformation in hybrid larch.

**Influence of the concentration of the infection solution and infection time on transformation.** The concentration of the infection solution and the infection time have a certain influence on the efficiency of *Agrobacterium*-mediated genetic transformation. When the infection time was 10~30 min, the transformation rate first increased and then decreased as the $OD_{600}$ value increased (Fig 7). When the infection time was 40 min and the infection solution $OD_{600}$ was 0.4 or 0.5, less resistant callus was obtained, and the $OD_{600}$ was 0.6 or 0.8, no resistant callus was obtained, which may be due to the *Agrobacterium* overgrowth producing poisoning, embryonic callus was difficult to sterilize, resulting in dying. When the infection time was 20~40 min and $OD_{600} \geq 0.5$, the transformation rate decreased as the $OD_{600}$ value increased. The transformation rate was the highest (45.56%) when the $OD_{600}$ value of the infection solution was 0.5 and the infection time was 20 min, there was a significant difference compared with other concentrations.

**Influence of the co-culture time on transformation.** It is generally believed that the process of exogenous gene transfer and integration into the recipient cell's genomic DNA occurs during the co-culture period, which takes at least 16 hours. As shown in Fig 6, different co-culture time had a certain impact on the conversion effect. In this study, calli transformed under the same conditions were cultured for 1, 2, 3, and 4 days. The results showed that a few resistant calli were obtained when co-cultured for 1 day (Fig 8A). Over a certain range, the transformation rate increased as the co-culture time increased, especially in the co-culture for 2 days (Fig 8B), the transformation rate was the highest at 37.03%, which compared with other co culture time, it showed obvious difference, and we found *Agrobacterium* appeared at the perifery of callus at this time (Fig 8B), it can be used as a time point for sterilization. However, when co-cultured for 3 days (Fig 8C), the result was the same with 1 day, and the transformation rate was decreased compared with 2 days (Fig 8D). Especially, when co-cultured for 4 days, the callus was completely buried due to *Agrobacterium* overgrowth given the long-term co-culture (Fig 8D), which made it difficult to sterilization, and even if *Agrobacterium* was inhibited during the recovery culture, the growth of embryogenic callus was also severely poisoned by *Agrobacterium* and could not resume normal growth. Eventually, we failed to obtain resistant callus, and the transformation rate was zero (Fig 8E).

**Obtained and validated resistant callus.** After the co-culture, numerous *Agrobacterium* were appeared on the surface of embryogenic callus. To not affect the subsequent growth of

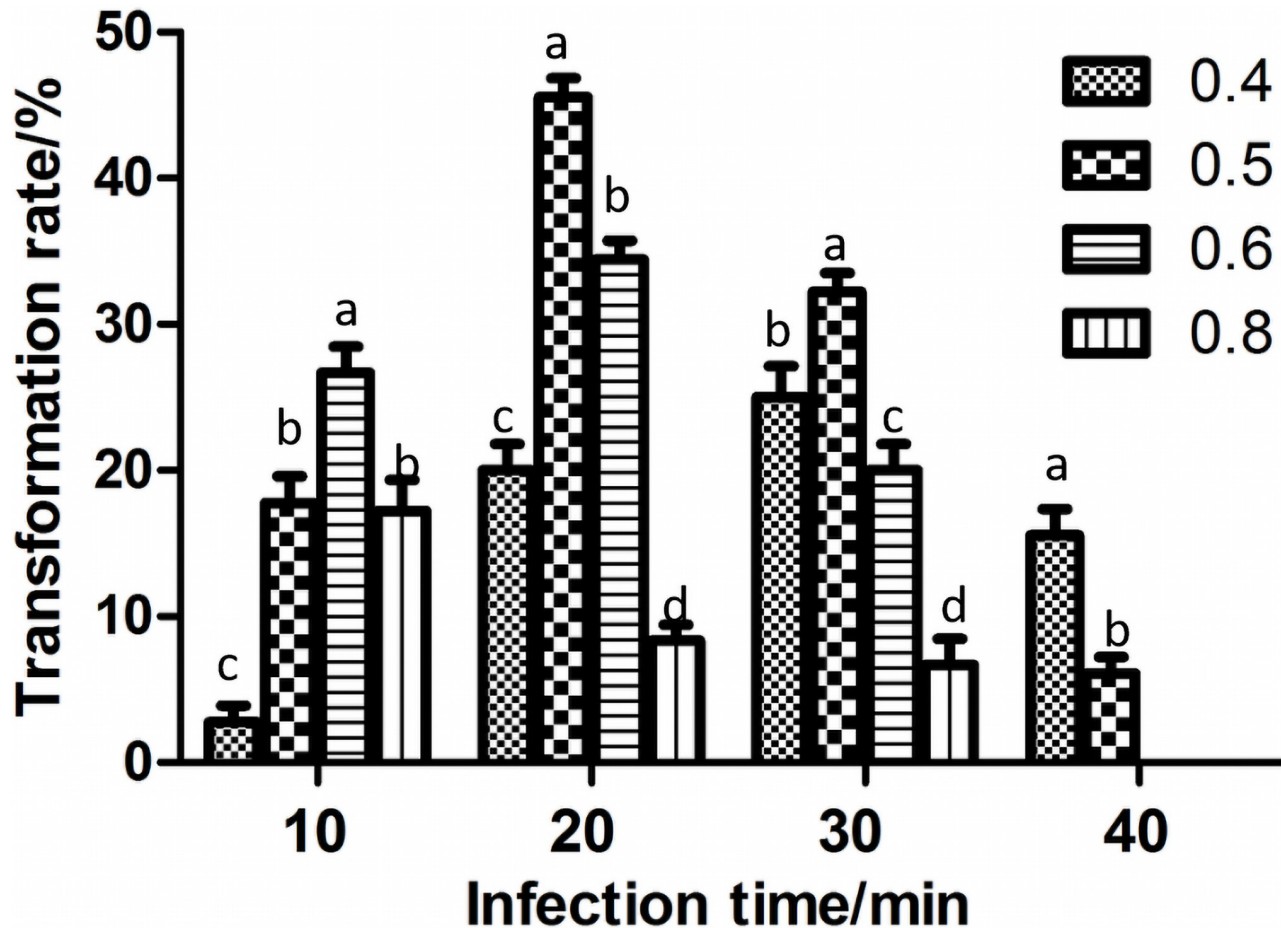

**Fig 7. Influence of the infection concentration and infection time on the transformation rate.** The $OD_{600}$ values of the infection solution concentration were 0.4, 0.5, 0.6 and 0.8. Different letters indicate significant differences at 0.05 level.

the callus, embryogenic callus needs to be cleaned and sterilized to eliminate *Agrobacterium* contamination [30]. First, the sample was washed twice with sterile water and then sterilizated twice on BM3 medium containing 500 mg/L cef for 5 min each time. After sterilizating, the sample was placed in recovery medium BM4 and cultured for one week. In addition, the sample was placed on screening medium BM5 containing hyg for screening culture (9 clusters per dish, 9 dishes in total) three times for three weeks each time (as shown in Fig 9); finally, 26 resistant calli were obtained, and the resistance callus rate was 32.1%. Regarding the resistant callus screened three times on the screening medium, three clusters were randomly selected, and after two weeks of recovery culture, PCR detection was performed. The results showed that a specific single band approximately 1026 bp in length was amplified from the transgenic callus DNA as a template (Fig 10A, the original blot and gel image was appeared in S1 File); wild-type callus DNA as a template was not amplified in the band, which was consistent with the expected results. The preliminary results indicated that the exogenous gene *hpt* from the empty vector was successfully inserted into the genome of embryogenic callus of hybrid larch. Furthermore, qRT-PCR analysis of transgenic and wild-type calli was carried out. The results showed that the different resistant callus with different expression levels, and were higher than the wild-type calli, which was also consistent with the expected results (Fig 10B). By GUS

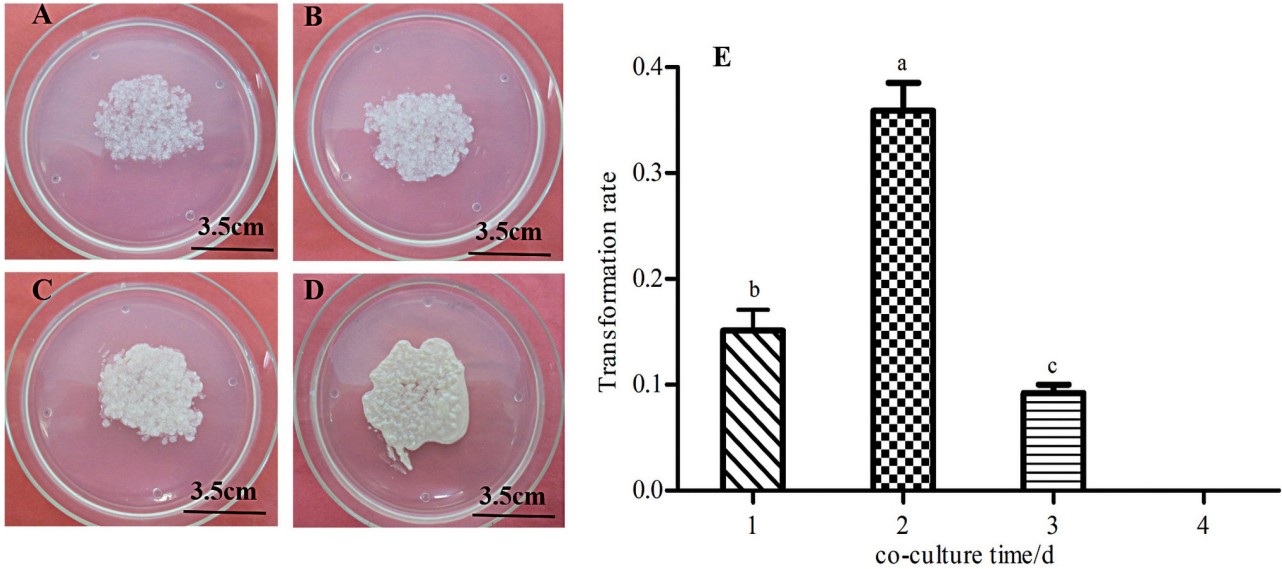

**Fig 8. Calli under different co-culture time and influence of the co-culture time on the transformation rate.** A~D: Calli in co-culture for 1, 2, 3, and 4 days. E: The transformation rate of calli in different co-culture time, different letters indicate significant differences at 0.05 level.

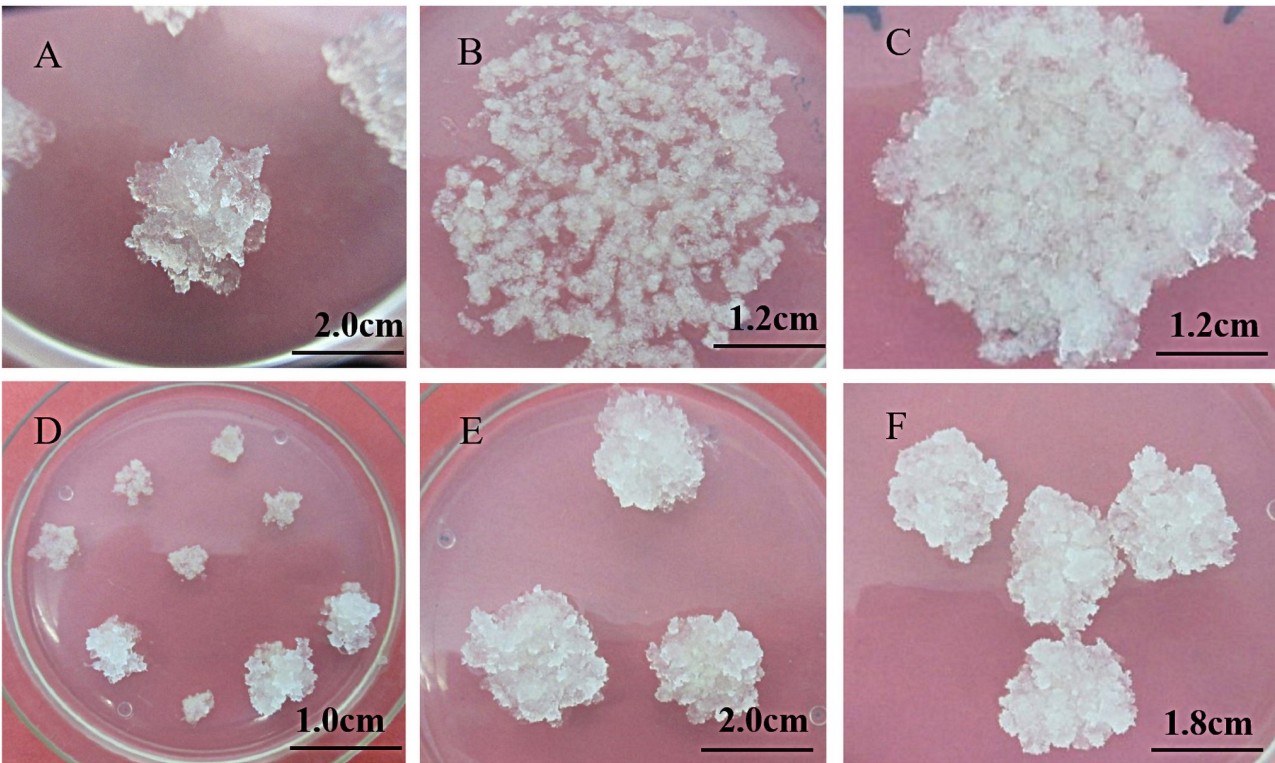

**Fig 9. *Agrobacterium*-mediated transformation of embryogenic callus and screening and validation of resistant callus.** A: Fresh embryogenic callus; B: Co-culture; C: Recovery culture; D, E, F: 1st, 2nd, and 3th screening culture.

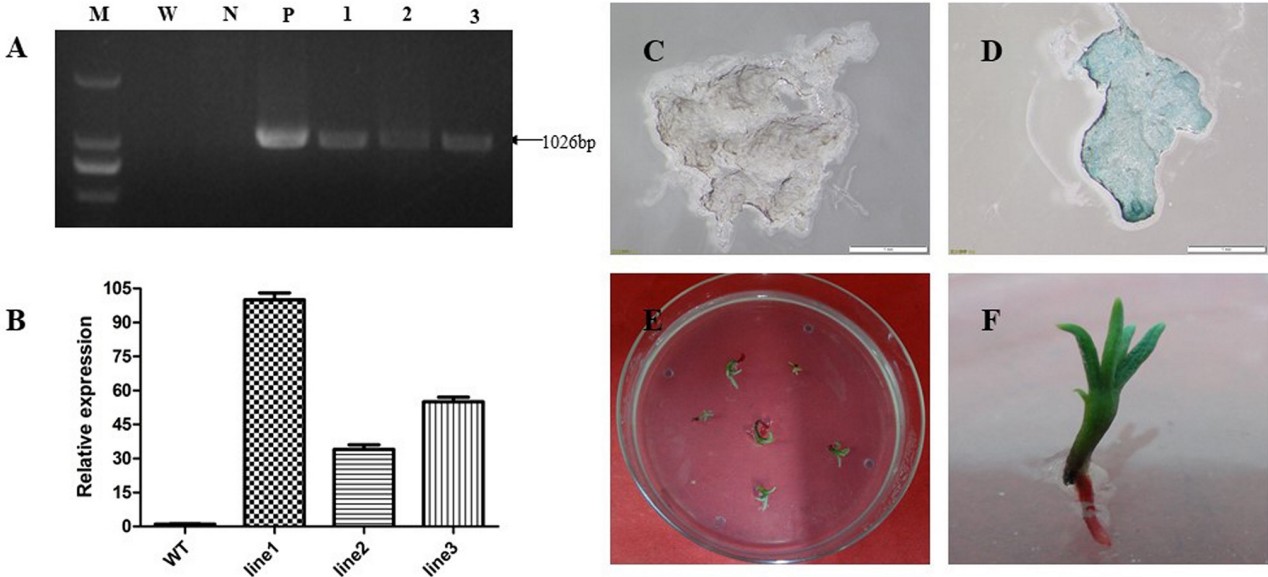

**Fig 10. Validation of resistant callus and somatic embryo seedling obtained.** A: The electropherogram of the PCR products contains: M: *GL* DNA Marker 2000; W: Water; N: Wild type callus; P: Empty vector plasmid; 1, 2, 3: Different transgenic empty vector callus; B: Relative expression of resistant callus by qPCR. line1, line2 and line3 refers to different transgenic empty vector callus line; C: Wild type callus GUS stain; D: Transgenic empty vector callus GUS stain; E: Somatic embryo germination; F: Somatic embryo seedling.

staining, wild type callus was not stained, but the calli transformed were stained blue completely (Fig 10C and 10D). Further, somatic embryogenesis on the obtained resistant calli were carried out according to germination of somatic embryos methods mentioned in the front part, and finally obtained germinating cotyledons embryo and somatic embryonic seedlings (Fig 10E and 10F). In this study, the transformation rate of resistance callus was 32.1%, which was 16~32 times higher than that reported by Levee. et al, who performed hybrid larch transformation experiments, the transformation rate was 1%~2% [23]. The application effect of the genetic transformation technology systems in different larch cell lines is good, and this study can provide a scientific basis for the genetic transformation of other larch and conifers.

## Discussion

Appropriate explant selection is essential for the induction of coniferous embryogenic callus during the induction of coniferous embryonic cells [31]. The effects of explant sources on the induction of embryogenic tissue were summarized in conifers and found that most conifer species use immature female gametophytes or zygotic embryos as explants to induce embryogenic tissues, which often leads to a higher induction. In contrast to other woody plants, the induction of embryogenic tissue in coniferous trees is strictly dependent on young tissues. It may be due to the difficulty in re-orientating coniferous species in vitro rapid propagation [28]. Embryogenic callus was first induced in 1987 in the female gametophyte of larch [32]. The development status of explant is important for induction, such as, the early embryo induction rate of cotyledons was found high in larch in the western United States, while immature zygotic embryos were easier to induce than mature zygotic embryos [5], in this study, immature zygotic embryos also used as the induction explant, and was induced successfully. Howe, the cotyledon stage zygotic embryos used for embryogenic callus induction in Japanese and European hybrid larch, and it was proved difficult [33]. Observations in many *Pinus* species

have suggested that zygotic embryos have the potential to produce embryogenic tissue at a very early development stage [34], and the same study results were found in *Pinus taeda* [35]. The developmental stages of zygotic embryos differed with different seed collection time, and different seed collection time can reflect the development stage. Simarly to above studies, in this study, immature zygotic embryos were used to induce embryogenic callus, and immature zygotic embryos collected at different stages were found to have different induction rates. The results showed that (Table 5) in the cones collected on June 27, which about 58 days after pollination and July 7, the embryogenic callus induction rate was low, i.e., 1.67% and 1%, respectively, and their embryogenicity was not easy to maintain in subsequent stages, resulting in death. It was difficult to induce embryogenic callus from cones collected on July 16, and the induction rate was 0. It is possible that the zygotic embryo had reached maturity, the differentiation ability was low, and the efficiency of induction was also reduced. The cones collected between July 1 and July 4 had a higher induction rate, and the induction rate of explants isolated from cones on July 1 was the highest, at 10.33%, and compared with other periods, there are significant differences by variance analysis. The results of analysis of variance showed that the collection time had a significant effect on the induction rate ($p = 0.000$). The developmental stage of the zygotic embryo in these two periods was the early cotyledonary (pre-cotyledonary) embryo, indicating that the best collection time for larch cones was the early cotyledonary stage, which were about 62 and 65 days after pollination. This finding is also consistent with the results reported in previous studies [36]. Therefore, selecting the appropriate seed collection time, ie. developmental stage of zygotic embryos at early cotyledonary (pre-cotyledonary) embryo, is necessary to inducing embryogenic callus.

Plant growth regulators also affect the induction of embryogenic callus. During the embryogenic process of cells, hormones regulate the whole life process of plant cell differentiation, growth direction and progress. Therefore, adding to different varieties and concentrations of exogenous hormones to the medium is critical for inducing embryogenic callus. Auxins, such as 2,4-D and NAA, cytokinins, such as 6-BA and KT, and added to can promote inducing embryogenic callus, which is commonly used in conifers [10]. In addition, some studies have suggested that 2,4-D is an important condition for the expression of larch embryonic competence, an indispensable factor in embryogenic callus induction [37]. For instance, 2,4-D had obvious effects on the induction of embryogenic callus of *Larix principis-rupprechtii*. When 2,4-D was higher or lower than 1.29 mg/L, it was not conducive for the induction of embryogenic callus [38], which is the similar to this study that 2,4-D lower or higher 1.0 mg/L, induce rate of embryogenic callus was increased. Studying the somatic embryogenesis pattern of Siberian larch, embryogenic callus obtained a higher induction rate when 2 mg/L 2,4-D and 0.5~1 mg/L 6-BA were added in medium [39] and 9 μM 2,4-D and 2.2 μM 6-BA [40]. It has also been reported that lower concentrations of growth regulators can increase the induction rate of *Pinus strobus* Linn. embryogenic callus from 20% to 53% [41]. Similar to the above

**Table 5. Effect of the cone collection date on embryogenic callus induction of hybrid larch.**

| Collection date | Development stage | Inoculation number | Induction rate (%) |
|---|---|---|---|
| June 27 | Column to pre-cotyledonary embryos | 300 | 1.67±1 bc |
| July 1 | Pre-cotyledonary embryos | 300 | 10.33±3.67 a |
| July 4 | Pre-cotyledonary embryos | 300 | 4.33±1 b |
| July 7 | Cotyledonary embryos | 300 | 1±1 bc |
| July 16 | Mature zytotic embryos | 300 | 0 c |

Notes: The values are mean±SD, and different letters indicate significant differences at 0.05 level.

results, in this study, the results showed that lower (0.2 mg/L) or higher (2.5 mg/L) 2,4-D concentrations could not induce embryogenic callus and that the induction rate increased a the 2,4-D concentration increased. The induction rate reached a maximum when the concentration of 2,4-D was 1.0 mg/L. In the induction of embryogenic callus in larch, the optimal hormone combination was 1.0 mg/L 2,4-D and 0.2 mg/L KT, which was consistent with previous studies; the auxin concentrations were higher than KT. However, this study differed from previous studies, and the induction rate with the combination of 2,4-D and KT was higher than that with 2,4-D and BA. It may be that the combination of 2,4-D and KT was more suitable for the induction of embryogenic callus in hybrid larch, *Larix kaempferi* 3× *Larix gmelinii* 9.

Subculture and regeneration of embryogenic callus are important components of somatic embryogenesis. In the process of proliferation, one of the most important phenomena is the accumulation of storage substances, such as protein and starch; as storage material accumulates, the embryonic cell cluster increases [42]. In the long-term subculture process, a high concentration of 2,4-D easily causes the loss of somatic embryogenesis. Therefore, in the propagation of embryogenic callus, 2,4-D should decrease over time or be replaced by NAA to benefit somatic embryogenesis [43]. In the proliferation and subculture process of embryogenic callus, many researchers reduced the plant growth regulator concentrations until they were suitable for maintaining proliferation and embryogenicity. Since plant cells at different developmental stages have different sensitivities to hormones, it is necessary to regulate the concentration of growth regulators during the process of embryogenic callus proliferation [44]. Therefore, we decrease 2,4-D and KT concentration for proliferation culture, and the results indicated that a 2,4-D concentration too high or too low in the medium of embryogenic callus was not conducive for the proliferation of embryogenic callus in hybrid larch. Following several treatments, embryogenic callus of hybrid larch, which was propagated on BM medium containing 2,4-D 0.5 mg/L and KT 0.2 mg/L for 12 days, proliferated the fastest, and the proliferation rate was also high at this time, reaching 323.08%. Because of high proliferation rate and its strong division ability, so it can also speed up subsequent experiments.

*Agrobacterium tumefaciens* was discovered can infect conifers, and studies have shown that coniferylol extracted from pine can promote conifers infection by *Agrobacterium tumefaciens* [17], similarly, studies also showed that the conifers are indeed the natural host of *Agrobacterium* [18]. However, it is affected by many factors [27]. Cell division ability has a significant effect on the transformation efficiency, and exogenous DNA is more easily integrated into cells in a dividing state [45]. Embryogenic callus in suspension culture with a strong cleavage ability as the acceptor material, which can effectively improve the genetic transformation efficiency of white spruce [46].

A prerequisite for successful genetic transformation is using high regeneration ability plant material, such as rapidly growing embryogenic tissue producing early somatic embryos with the ability to regenerate somatic seedlings. Charity pre-cultured cotyledonary for 0~10 days in *Pinus radiata* before *Agrobacterium* infection and the transformation rate of the 0~3 days preculture was 5~20%, and pre-culturing for 6~9 days showed the highest transformation rate, at 55%. It may be that the former was in the preparation period of cell division, while the latter was in the vigorous period of cell division [47]. Studying the genetic transformation of *Pinus pinaster* Ait., the results showed that material pre-cultured for 4 days in suspension medium could be successfully transformed, but pre-culturing for 7 days was unsuccessful; material pre-cultured for 4 days was likely in a period of high cell division, which can improve the conversion efficiency [48]. Therefore, selecting cells with a stronger division ability and rapidly growing embryogenic tissue as the acceptor material for genetic transformation is a basic condition for successful transformation. In this study, when embryogenic callus was subcultured, it was found that 12 days was the best subculture time, which was also a strong divisional state.

Therefore, embryogenic callus subcultured for 12 days was selected as the acceptor material, and a high transformation efficiency was achieved. The embryogenic tissue of larch is sensitive to *Agrobacterium*, genetic transformation of exogenous genes is actually a process in which strains interact with exogenous genes. The *Agrobacterium* load and infection time are also important determinants for successful transformation [49]. Therefore, the transformation efficiency can be enhanced within a certain range to enhance the inoculation intensity of *Agrobacterium*, including the infection concentration and sterilization and co-culture time. The process of infection is mainly *Agrobacterium* attaching to the receptor material. If the concentration of the infection solution is too low or the infection time is too short, the transformation efficiency will be reduced due to insufficient adhesion of *Agrobacterium*. Similarly, if the concentration of the infection solution is higher or the infection time is longer, *Agrobacterium* will be excessively attached, which will result in incomplete sterilization; the receptor material survival rate will decrease, and the conversion efficiency will be reduced [50,51]. When the infection solution $OD_{600} = 1$, the receptor material died due to the poisoning effect of the bacterial liquid [51]. In the ideal genetic transformation of conifers, $OD_{600}$ is generally controlled between 0.4 and 0.6 and the infection time is controlled between 10~30 min. For Japanese larch, when the $OD_{600}$ was 0.4 and the infection time was 10~20 min, the transformation efficiency was the best [52]. In this study, the results were consistent with the above results. When the infection time was 40 min and the the infection solution $OD_{600}$ was 0.6 and 0.8, no resistant callus was produced. Even if the $OD_{600}$ was 0.4, only less resistant callus was produced. Similarly, even if the infection solution $OD_{600}$ was 0.8 and the infection time was 10 min, only less resistant callus was produced. When the $OD_{600}$ of the infection solution was 0.5, the transformation efficiency was the highest at 45.56%. However, studies have shown that the co-culture time is the most important factor in genetic transformation [27] because co-culturing is the process of completing genomic transfer of *Agrobacterium tumefaciens* T-DNA to the receptor materials. Co-culture for 2~3 days is necessary for ensuring complete transfer of the T-DNA from *A. tumefaciens* to plant cells. If the co-culture time is short, it may not be long enough to complete the T-DNA transfer. If the T-DNA transfer takes a long time, it may lead to excessive growth of *Agrobacterium*, and can considerably lower plant viability and reduce the transformation. The growth state of *Agrobacterium* depended on the concentration of the infection solution and infection time in the co-culture stage, and a higher conversion rate was obtained when the co-culture was approximately 2 days, which was proposed by Zhu [52]. The highest transformation rate (11.4 events per gram of fresh mass) was obtained when a vigorously growing embryonal mass had an $OD_{600}$ of 0.3 in the infection solution and was co-cultured for 72 h [53]. The optimum transformation efficiency in this study was obtained with a moderate bacterial load and co-culturing for 2 days, which is similar to previous findings [36,52]. In this study, when cultured for 1 or 3 days, a few resistant calli were obtained. The transformation rate was 0 when the co-culture was 4 days. Because a large amount of *Agrobacterium* adhered to the surface of the callus, it was very difficult to sterilize; thus, eventually, they all died. When the callus was co-cultured for 2 days, the transformation rate was the highest, at 35.65%. In the study, the results also showed that resistant calli were obtained after the first screening, and they did not die after the second and third screenings, i.e., which suggested the false positive rate was low. Therefore, determining the optimal co-culture time, adjusting the activity of *Agrobacterium* and reducing its damage to the receptor material are very important for the genetic transformation of larch, and it is necessary to observe the growth of *Agrobacterium* every day during co-culture. It is essential to terminate the co-culture at the right time to achieve a higher transformation rate. In addtion, the transformation ability of *Agrobacterium* strains to the recipient material is different, strains EHA105, LBA4404, and GV3850 with same plasmid with GUS tags were used for infecting black sprucce cotyledons, results

showed that GUS expression level was with EHA105 higher than the other two strains [52]. In addition, GV3101 stain was used as trasformation stain, and the genetic transformation of the embryogenic callus of *Larix olgensis* was successfully realized [54]. So, in this study, GV3101 stain was selected for infecting embryogenic callus in hybrid larch *Larix kaempferi* 3x *Larix gmelinii* 9. In actual experiments, it is also necessary to screen strains based on recipient mate-rial materials. This study provides a basis for subsequent genetic function studies and the genetic transformation of other larch and conifers.

## Conclusions

Embryogenic callus induction was studied using a combination of different plant growth regu-lators and different concentrations. The highest induction rate of 10.83% was observed in embryogenic callus cultured in BM medium with 1.0 mg/L 2,4-D (2,4-dichlorophenoxyacetic acid) and 0.2 mg/L KT (kinetin). When cultured on a proliferation medium for 12 days, prolif-eration was the fastest, which could not only maintain the freshness and vitality but also resulted in a large amount of proliferation. The suitable pre-culture medium for somatic embryogenesis was 1/4 BM medium containing 10 g/L inositol and 60 g/L sucrose. The combi-nation of 45 mg/L ABA and 75 g/L $PEG_{4000}$ could promote the number of somatic embryos, and reached the maximum. In addtion, the transformation rate of resistance callus can reach 32.1% under the condition that optimal $OD_{600}$ of the infection solution $OD_{600}$ was 0.5, co-cul-ture time was 2 days and concentration of hyg was 4 mg/L, which was 16~32 times higher than that of the previous genetic transformation of Japanese European Hybrid Larch (1%~2%). This study could be beneficial for the innovation and breeding of larch germplasm resources by genetic engineering and other means and provides a certain basis for the preservation and rapid propagation of excellent larch germplasm resources and genetic engineering breeding and genetic function studies of larch and other conifers.

## Materials and methods

### Plant materials

The plant material, the immature cones of hybrid larch *Larix kaempferi* 3x *Larix gmelinii* 9 freely pollinatedis(ultivated and open pollination in an orchard) without damaging the plant population and comply with relevant institutional, national, and international guidelines and legislation, selected from seedling seed orchard, were collected on June 27, July 1, July 4, July 7, and July 16, 2017(about 58, 62, 65, 68 and 77 days after pollination, respectively), from the Qingshan seed orchard, Linkou County, Heilongjiang Province and the plant materials in this study are available from the Qingshan seed orchard upon reasonable request. The collected cones were packed and marked in self-sealed bags, stored in ice boxes and taken to the labora-tory. Then, the cones were stored in a 4 ˚C refrigerator until use within 14 days.

### Disinfection and inoculation of explants

The cones were selected with fully filled growth, no pests and good development status; the seeds were pulled from the cones; the seed coat was peeled; the samples were refrigerated in a 4 ˚C refrigerator for 12 h, and then, the following disinfection treatment was applied: first, the explant was washed with sterile deionized water 3 times, followed by 75% alcohol for 1 min, and cleaned with sterile deionized water 3 times; finally, the explant was disinfected with 3% NaClO for 8 min, 10 min, 12 min, 15 min, 18 min, and 20 min and cleaned with sterile deion-ized water 3 times.

After the disinfection, the water on the surface of the explant was blotted dry with sterile filter paper. The following two disinfection methods were carried out:(1) the seeds were cut longitudinally, and the embryo and downward incised endosperm were inoculated in embryogenic callus induction medium, or (2) an incision was made from the top of the seed, and the embryo was removed from the endosperm, placed on sterile filter paper until the water on the surface was blotted dry, and finally, inoculated in embryogenic callus induction medium.

## Induction of embryogenic callus

The sterilized explants were inoculated on BM basic medium. Four levels of plant growth regulator 2, 4-D (0.2, 1.0, 1.8, and 2.5 mg/L), 6-BA (0, 0.2, 0.5, and 1.0 mg/L), and KT (0, 0.2, 0.5, and 1.0 mg/L) were added, and an $L_{16}(4^5)$ orthogonal experimental design was used to determine the optimal conditions [55]; additives: inositol 1 g/L, glutamine (Gln) 0.45 g/L, acid hydrolysed casein (CH) 0.5 g/L; sucrose 30 g/L; and agar 6 g/L. The the medium pH was adjusted to 5.8 with 1 mol/L KOH and HCl before autoclaving at 121 ˚C for 21 min. Each petri dish was inoculated with 5~10 immature zygotic embryos in embryogenic callus induction medium, and each combination was subjected to 3 biological replicates. After the inoculation, embryogenic callus was induced in a dark culture chamber (23±2˚C), and its state was observed after one week. The induction rate was calculated after 1~2 months.

## Proliferation of embryogenic callus and determination of subculture time

Proliferation of embryogenic callus: After subculturing the embryogenic calli three times (3 weeks each time) (no browning or other instable phenomena) in induction medium BM, the calli were propagated and subcultured in proliferation medium. The plant growth regulator 2,4-D was reduced to 1/2, 1/5, and 1/10, and KT was reduced to 1/2, 1/5, and 1/10 in the proliferation medium (Table 6). The additives sucrose and agar are the same as the BM induction medium.

Fresh embryogenic callus was taken and using Carbol fuchsin staining solution for cytological observation, staining time was 8~10 min, then observed the structure of embryogenic callus with an optical microscope and took pictures. The nucleus structure of the embryonic substance was stained red by carbol fuchsin and the embryo stalk structure was not stained red.

Determination of the subculture time: The initial inoculation amount of 130 mg/cluster embryogenic callus was cultured in the proliferation medium as follows: three dishes per medium, and 3 clusters per dish. The fresh weight of the callus was weighed every 3 days (a

**Table 6. Proliferation medium for embryogenic callus.**

| Proliferation medium | Plant growth regulator and concentration (mg/L) | |
|:---:|:---:|:---:|
| | **2,4-D** | **KT** |
| 1/2- BM | 0.5 | 0.2 |
| 1/2↓ BM | 0.5 | 0.1 |
| 1/5- BM | 0.2 | 0.2 |
| 1/5↓ BM | 0.2 | 0.1 |
| 1/10- BM | 0.1 | 0.2 |
| 1/10↓ BM | 0.1 | 0.1 |

Notes: 1/2-BM represented that 2,4-D was reduced to half of the 2,4-D concentration in BM medium, KT remained unchanged. 1/2↓BM represented that 2,4-D and BA were reduced to half of the 2,4-D and BA concentration in BM medium. 1/5-BM, 1/5↓ BM, 1/10-BM and 1/10↓ BM were Similarly.

sterile empty culture dish was weighted; then, the callus was placed in the sterile culture dish, and the difference in weight was calculated) until the callus grew into a recession; the samples were weighed for 45 days continuously. The growth curve of embryogenic callus was generated, and the optimal subculture time and optimal proliferation medium were determined.

## Somatic embryo maturation and germination

Pre-culture of embryogenic callus: pick 200 mg/cluster of fresh embryogenic callus grown on the proliferation medium for 10 days as initial inoculation amount, and removed the plant growth regulator, added 30 g/L, 60 g/L and 90 g/L sucrose, inositol 1, 10 and 15 g/L, agar 6.5 g/L, glutamine 0.45 g/L, hydrolyzed casein 0.5 g/L. In order to achieve synchronization of embryogenesis and higher occurrence. BM modified medium, which medium ion was in 1/4, 1/2 and 1 times (ie 1/4 BM, 1/2 BM and BM) BM culture respectively, used for pre-culture (transition culture) for 10 days before somatic embryo induction. The experiment adopts the orthogonal design of $L_9(3^4)$, and carries out dark culture at 23±2˚Cto realize the transitional culture of somatic embryos. The above treatments were repeated 3 times, and 3 cluster embryogenic callus were repeatedly inoculated each time.

Induction of somatic embryos: After transitional culture of callus, the whole mass was then transferred to BM somatic embryo induction medium without 2,4-D and KT, supplemented with 15, 30, 45, 60 mg/L ABA, 0, 50, 75 and 100 g/L $PEG_{4000}$, 1 g/L inositol, 0.675 g/L glutamine, 0.25 g/L acid hydrolyzed casein, 60 g/L sucrose, 6.5 g/L agar. The above treatments were repeated 3 times, and 3 cluster embryogenic callus were repeatedly inoculated each time.

Germination of somatic embryos: After dark culture for 4 to 6 weeks, 50 cotyledon embryos that grew normally close to zygotic embryos were selected and subjected to the filter paper bridge method "drying" treatment [56], and then placed in WPM medium without plant growth regulators, and 1 g/L activated carbon, 10 g/L sucrose concentration and 6.5 g/L agar were added for germination, and then placed it in a light culture room (23±2˚C, 16 h/d light) for light culture. The germination rate was counted after 2 weeks.

## Genetic transformation of hybrid larch

**Materials.** The fresh embryogenic callus cultured on proliferation medium for 10 days was selected as the genetically transformed material.

A pCAMBIA1301 vector containing the hyg marker gene *hpt* was used for genetic transformation in hybrid larch, and the pCAMBIA1301 plasmid was generated at our laboratory. The DH5a strains (TRANSGEN BIOTECH, Beijing, China) and GV3101 strains (TRANSGEN BIOTECH, Beijing, China) were used for *E.coli* transformation and *Agrobacterium* transformation, respectively.

**Culture medium used in the experiment.** The medium used in this experiment is shown in Table 7. All media were autoclaved at 121 ˚C for 21 min, and once the temperature of the media was lower than 50 ˚C, glutamine(Gln), acid hydrolysed casein(CH), inositol, kanamycin (Kan), gentamicin(Gen), cefotaxime(Cef), hygromycin B(Hyg), Acetosyringone (As), 2 4-D and KT were added after filtering.

**Sensitivity experiment of antibiotics.** Hygromycin has a greater toxic effect on receptor material and can play a significant role at a low concentration and over a short period of time; thus, it has a good screening effect. Trontin et al. showed that the resistant tissues obtained by hygromycin screening were positive by PCR [24], while kanamycin screening showed a nearly 90% false positive rate [25,26]. In this study, 400 mg/cluster embryogenic callus on fresh proliferation medium was selected and placed on BM0 medium containing 0, 2, 4, 6, or 8 mg/L hyg and 0, 10, 20, 30, or 40 mg/L kan for 21 days. The fresh weight of embryogenic callus was

**Table 7. Culture media used in the genetic transformation of embryonic callus.**

| Medium | Medium Composition | Function |
|---|---|---|
| LB0 | NaCl10 g/L+Tryptone10 g/L+Yeast 5 g/L | Liquid culture of bacteria |
| Solid LB0 | LB0+Agar12 g/L | Solid culture of bacteria |
| LB1 | LB0+Kan50 mg/L | *E. coli* culture |
| LB2 | LB0+Kan50 mg/L+Gen50 mg/L+Rif 50 mg/L | *Agrobacterium* culture |
| BM0 | BM+Sucrose 30 g/L+CH 0.5 g/L+Gln 0.45 g/L+Inositol1 g/L+2,4-D 0.1 mg/L +KT 0.2 mg/L, PH 5.8 | Suspension liquid |
| BM0 Solid | BM+Sucrose 30 g/L+Agar 6.5 g/L+CH 0.5 g/L+Gln 0.45 g/L+Inositol1 g/L +2,4-D 0.1 mg/L+KT 0.2 mg/L, PH 5.8 | Proliferation culture |
| BM1 | BM0+As 100 μM | Infection solution |
| BM2 | BM1+Agar 6.5 g/L | Co-culture |
| BM3 | BM0+Cef 500 mg/L | Degerming liquid |
| BM4 | BM0 Solid+Cef 200 mg/L | Recovery culture |
| BM5 | BM4+Hyg 4 mg/L | Screening culture |

obtained every 3 days, and its growth status was observed. Finally, the best antibiotic and concentration were determined.

**Genetic transformation, attainment and detection of resistant callus.** The bacteria were cultured in LB liquid medium containing kan, gen and rif, and an overnight culture was performed prior to the following infection test. When the *Agrobacterium* liquid $OD_{600}$ reached 0.8~1.0, the liquid was centrifuged at 4 ˚C at 8000 g for 15 min. The supernatant was completely discarded, and the cell pellet was resuspended in the BM0 suspension, i.e., the "bacterial weight suspension method" was used to prepare an "inoculation solution", BM1. Then, 2 g/cluster fresh embryogenic callus crushed and separated with tweezers on proliferation medium were selected and placed in the prepared infection solution BM1 for infection under gentle shaking during this period. The infected callus was adsorbed with excess liquid on the surface by sterilized filter paper and placed in BM2 to co-culture in the dark.

In this study, the following factors were tested to determine whether they affect the genetic transformation efficiency of hybrid larch:

1. $OD_{600}$ of infection solution BM1: 0.4, 0.5, 0.6, and 0.8;

2. Infection time: 10 min, 20 min, 30 min, and 40 min;

3. Co-culture time: 1, 2, 3, and 4 days.

To validate the transgenic callus, genomic DNA from wild-type and transgenic embryogenic callus was extracted by a plant genomic DNA extraction kit (TIANGEN BIOTECH, Beijing, China). DNA was used as the template, primers were designed based on the hyg gene sequence of the vector. The primers sequences used for validation at the DNA level were as follows: *Hyg*-F: 5'-ATGAAAAAGCCTGAACTCACCGCGA-3'; *Hyg*-R: 5'-CTATTTCTTTGCCCTC GGACGAGTG-3', and PCR was performed. Meanwhile, total RNA was extracted by an RNA extraction kit (BioTeke, Beijing, China) and reverse transcribed into cDNA by a reverse transcription kit (Takara, Beijing, China); then, qRT-PCR reaction was performed using a qRT-PCR kit (TransGen Biotech, Beijing, China) as described by Zhang [57], which was used for expression analysis of the transgenic and wild-type callus. Using *EF-1* as the internal control gene, which has been screened and verified [58], the gene expression levels were

normalized. *EF-1*-F: 5'-GACTGTACCTGTTGGTCGTG-3'; *EF-1*-R: 5'- CCTCCAGCAGAGCT TCAT-3'. The quantitative primer was as follows according to the hyg gene sequence of the vector: upstream primer: 5'-GGGCGAAGAATCTCGTGCT-3'; downstream primer: 5'-CTCC CCAATGTCAAGCACTTC-3'. The initial data of qRT-PCR were processed with Excel 97–2003, and graphed with GraphPad Prism5 software. The relative expression of transgenic callus and wild callus was quantitatively analysed using the $2^{-\Delta\Delta Ct}$ method ($\Delta Ct$ = the differential value from Ct average of target gene and internal reference gene; $\Delta Ct$ = the differential value of test sample $\Delta Ct$ and control sample $\Delta Ct$ [59]. In addition, add the resistant callus into a clean centrifuge tube, add an appropriate amount of GUS staining solution to make all the materials invade the staining solution, and use a vacuum pump to vacuum for 5 min. Placed in a 37 ˚C constant temperature water bath for 3 days, observe the staining of resistant callus. In general, resistant tissues appear blue-green in the staining solution, but no-resistant tissues did not.

## Data statistics

Induction rate of embryogenic callus (%): Number of immature zygotic embryos of embryogenic callus obtained/number of immature zygotic embryos inoculated ×100%

Proliferation rate of embryogenic callus (%): Fresh weight of embryogenic callus after proliferation/fresh weight of embryogenic callus before proliferation ×100%

Somatic embryogenesis (number/g FW): Number of somatic embryogenesis/fresh weight of initial inoculation during pre-culture

Transformation rate of callus (%): Number of resistant callus/number of callus on screening medium ×100%

Data analysis was performed using IBM SPSS Statistics 19 software, and the percentages were transformed by arcsine to perform variance analysis, and Duncan method was used for multiple comparisons.

## Supporting information

**S1 Table. Media compound and composition content used in the text.**
(DOCX)

**S1 File. The original blot and gel image.**
(PDF)

## Author Contributions

**Data curation:** Sufang Zhang, Shanshan Yan, Chen Wang.

**Formal analysis:** Sufang Zhang.

**Methodology:** Hanguo Zhang.

**Validation:** Sufang Zhang, Peiqi An, Hanguo Zhang, Lei Zhang.

**Writing – original draft:** Sufang Zhang.

**Writing – review & editing:** Sufang Zhang, Qing Cao, Junhui Wang, Hanguo Zhang, Lei Zhang.

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
