## [Decision Letter · Decision Letter 0]

4 Jun 2021

PONE-D-21-16841

Embryogenic callus induction from immature zygotic embryos and genetic transformation of Larix kaempferi 3x Larix gmelinii 9

PLOS ONE

Dear Dr. Zhang,

Thank you for submitting your manuscript to PLOS ONE. After careful consideration, we feel that it has merit but does not fully meet PLOS ONE’s publication criteria as it currently stands. Therefore, we invite you to submit a revised version of the manuscript that addresses the points raised during the review process.

We look forward to receiving your revised manuscript.

Kind regards,

Vijay Kumar

Academic Editor

PLOS ONE

Journal Requirements:

2.PLOS ONE now requires that authors provide the original uncropped and unadjusted images underlying all blot or gel results reported in a submission’s figures or Supporting Information files. This policy and the journal’s other requirements for blot/gel reporting and figure preparation are described in detail at https://journals.plos.org/plosone/s/figures#loc-blot-and-gel-reporting-requirements and https://journals.plos.org/plosone/s/figures#loc-preparing-figures-from-image-files. When you submit your revised manuscript, please ensure that your figures adhere fully to these guidelines and provide the original underlying images for all blot or gel data reported in your submission. See the following link for instructions on providing the original image data: https://journals.plos.org/plosone/s/figures#loc-original-images-for-blots-and-gels.

Reviewers' comments:

Reviewer's Responses to Questions

**Comments to the Author**

1. Is the manuscript technically sound, and do the data support the conclusions?

Reviewer #1: Partly

2. Has the statistical analysis been performed appropriately and rigorously? 

Reviewer #1: No

3. Have the authors made all data underlying the findings in their manuscript fully available?

Reviewer #1: Yes

4. Is the manuscript presented in an intelligible fashion and written in standard English?

Reviewer #1: No

5. Review Comments to the Author

Reviewer #1: In this study, immature zygotic embryos of the hybrid larch Larix kaempferi 3 x Larix gmelinii 9 were used as explants, embryogenic callus induction and genetic transformation was claimed.

The manuscript is not presented in an intelligible fashion and written in standard English. In general the manuscript is wordy and truncated. English language/usage has to be carefully revised. It is truncated and very difficult to the reader to follow it

The Introduction does not clearly state the aim of the research and the present knowledge. It should be deeply revised.

Authors should revise the way of citing authors from the very Introduction. Line 81: Smith discovered ( ???? ); Line 88 : Sederoff applied (Sederoff et al.) ; Line 90: Huang obtained (* Huang et al. ) ; Line 91: Shin infected the hypocotyls (Shin et al.)…; Line 93: Levee used (Leéve at al.); just to mention a few.

By the way, what does ´Shin infected' really mean?

Line 26: Please define BM at first mention

Line 30: ABA should be defined at first mention

Line: PEG should´ve been defined at first

Lines 32 and 41 81,82, 84, 85, 86,87, 88, 92, 94 : Agrobacterium should appear in italics!

Line: What does “ transformation rate of resistance callus” mean? Resistant???

Line 52/53: It is suggested: Since embryos of L. x eurolepis and L.x leptoeuropaea at precotyledonary developmental stage were used as an....

Data on “ Optimal sterilisation conditions for explants” are so elementary and irrelevant to the scope of the manuscript. It should be briefly mentioned in Material and Methods. Note that they have not even been discussed.

References should be carefully standardized. On top of that some key references should be considered, moreover the recent publication by Song et al. 2020 should be really looked into:

Ismail G., Schnabl H., Zoglauer K., Boehm R. (2004). Agrobacterium-mediated transformation of Larix decidua: an assessment of factors influencing the efficiency of gus gene transfer. J. Appl. Bot. Food Qual. 78 83–90

Song Y., Zhang H., Li S., Li S. (2016). Relationship between the induction of embryogenic callus of larch and the morphology of immature embryos. J. North-East Forest. Univ. 44 25–30.

Song Y, Bai X, Dong S, Yang Y, Dong H, Wang N, Zhang H, Li S. Stable and Efficient Agrobacterium-Mediated Genetic Transformation of Larch Using Embryogenic Callus. Front Plant Sci. 2020 Nov 25;11:584492. doi: 10.3389/fpls.2020.584492.

Lines 76-78: Please delete it since it is way obvious the statement since it does not add any information: “ As is well known, the premise of plant somatic embryogenesis is that embryogenic callus is induced and can proliferate stably; thus, it is necessary to induce stably proliferating embryogenic callus in hybrid larch.

Line 330: After the co-culture, numerous Agrobacterium were symbiotic on the surface of embryogenic callus. Symbiotic, REALLY?

Line 858-859: What does ‘ E: cotyledons germinate’ mean?

Figures 1, 2, 4, 8 and 9 do lack size reference bars.

The Discussion sounds more like a Literature Review.... (see lines 357 to 372). It is poorly elaborated as presented.

Lines 358 and 362: Stasolla summarized ... Stasolla believe... Authors shouls sitck to a scientific writing style. Even though it should appear Satasolla ang Yeung [ ]

Line 372: … stage was the progeny stage [32]. What does progeny stage mean?

Lines 372-389: In what sense authors want to stress the collection dates/times? Does it really matter? In my opinion the stage is what really matters. Above all, authors do not provide a convincing explanation to justify this difference. The discussion is way superficial.

Line 386: This finding is also consistent with the results reported in previous studies. Please, specify them!

And this also applies to the dose-response data (lines 392-423) during induction step. The Discussion again sounds as a literature review. The finding are not really compared and contrasted with existing and referenced literature.

If one have a look from line 441-454 this very same style appears denoting a literature review on the topic (Agrobacterium-mediated transformation).

For transformation purposes why authors did not consider adding acetosyringone or coniferyl alcohol as vir gene inducers? How about the Agro strain GV 3101 used as compared to others - C58 /EHA 105? It would be interesting to consider for the Discussion.

Overall, the ms lacks concision and clearness, and it is just borderline. That said it is advocated that the manuscript should be carefully revised, and as presented does not merit to be published.

6. PLOS authors have the option to publish the peer review history of their article (what does this mean?). If published, this will include your full peer review and any attached files.

Reviewer #1: No

---

## [Author Response · Author response to Decision Letter 0]

13 Aug 2021

Dear editor:

Thanks to the editors and reviewers for valuable comments on the manuscript "Embryogenic callus induction from immature zygotic embryos and genetic transformation of Larix kaempferi 3 x Larix gmelinii 9". The author has made detailed revisions in accordance with the comments. If there are any inadequacies, please correct or point out them. The answers are as follows:

Author's responses to the editor

Answer: My manuscript meets PLOS ONE's style requirements, including those for file naming.

2.PLOS ONE now requires that authors provide the original uncropped and unadjusted images underlying all blot or gel results reported in a submission’s figures or Supporting Information files. This policy and the journal’s other requirements for blot/gel reporting and figure preparation are described in detail at https://journals.plos.org/plosone/s/figures#loc-blot-and-gel-reporting-requirements and https://journals.plos.org/plosone/s/figures#loc-preparing-figures-from-image-files. When you submit your revised manuscript, please ensure that your figures adhere fully to these guidelines and provide the original underlying images for all blot or gel data reported in your submission. See the following link for instructions on providing the original image data: https://journals.plos.org/plosone/s/figures#loc-original-images-for-blots-and-gels.

 Answer: I have resubmit the figures accroding to the Plos one requirement， and my blot/gel image data has been named ‘S1_raw_image’ and uploaded as a Supporting Information file. In my cover letter, “my blot/gel image data are in Supporting Information“ has been noted.

 Answer: My ethics statement have been moved to the Methods section of my manuscript.

Author's responses to the reviewers

Reviewer #1: In this study, immature zygotic embryos of the hybrid larch Larix kaempferi 3 x Larix gmelinii 9 were used as explants, embryogenic callus induction and genetic transformation was claimed.

(1) The manuscript is not presented in an intelligible fashion and written in standard English. In general the manuscript is wordy and truncated. English language/usage has to be carefully revised. It is truncated and very difficult to the reader to follow it

The Introduction does not clearly state the aim of the research and the present knowledge. It should be deeply revised.

Answer: English language has been revised in my manuscript.

 The ”Introduction” about the aim of the research and the present knowledge has been revised. 

(2) Authors should revise the way of citing authors from the very Introduction. Line 81: Smith discovered ( ???? ); Line 88 : Sederoff applied (Sederoff et al.) ; Line 90: Huang obtained (* Huang et al. ) ; Line 91: Shin infected the hypocotyls (Shin et al.)…; Line 93: Levee used (Leéve at al.); just to mention a few.

Answer: The way of citing authors have been revised from the very Introduction and other sections in this manuscript. 

(3) By the way, what does ´Shin infected' really mean?

Answer: ´Shin infected' has been revised as shown in this manuscript.

(4) Line 26: Please define BM at first mention

Answer: ”Supplement table 1” has been submited separately to define BM.

(5) Line 30: ABA should be defined at first mention

Answer: ”ABA” has been defined in this manuscript.

(6) Line: PEG should´ve been defined at first

Answer: ”PEG” has been defined in this manuscript.

(7) Lines 32 and 41 81,82, 84, 85, 86,87, 88, 92, 94 : Agrobacterium should appear in italics!

Answer: The“Agrobacterium” appeared in ”Lines 32 and 41 81,82, 84, 85, 86,87, 88, 92, 94 and other sections” have been revised in italics.

(8) Line: What does “ transformation rate of resistance callus” mean? Resistant???

Answer: ”transformation rate of resistance callus” has been replaced with ”transformation rate of callus”.

(9): Line 52/53: It is suggested: Since embryos of L. x eurolepis and L.x leptoeuropaea at precotyledonary developmental stage were used as an....

Answer: “ Since Klimaszewska used embryos of L.x eurolepis and L.x leptoeuropaea at precotyledonary developmental stage as an ....“ has been replaced with “Since embryos of L. x eurolepis and L.x leptoeuropaea at precotyledonary developmental stage were used as an....”

(10) Data on “ Optimal sterilisation conditions for explants” are so elementary and irrelevant to the scope of the manuscript. It should be briefly mentioned in Material and Methods. Note that they have not even been discussed.

Answer: “Optimal sterilisation conditions for explants” was correponding to the “Disinfection and inoculation of explants” in “material and methods” section, so this section was retained.

(11) References should be carefully standardized. On top of that some key references should be considered, moreover the recent publication by Song et al. 2020 should be really looked into:

Ismail G., Schnabl H., Zoglauer K., Boehm R. (2004). Agrobacterium-mediated transformation of Larix decidua: an assessment of factors influencing the efficiency of gus gene transfer. J. Appl. Bot. Food Qual. 78 83–90

Song Y., Zhang H., Li S., Li S. (2016). Relationship between the induction of embryogenic callus of larch and the morphology of immature embryos. J. North-East Forest. Univ. 44 25–30.

Song Y, Bai X, Dong S, Yang Y, Dong H, Wang N, Zhang H, Li S. Stable and Efficient Agrobacterium-Mediated Genetic Transformation of Larch Using Embryogenic Callus. Front Plant Sci. 2020 Nov 25;11:584492. doi: 10.3389/fpls.2020.584492.

Answer: References have been standardized, and the recent publication by Song et al. 2020 have been looked into and was some references have been added to this manscript.

(12) Lines 76-78: Please delete it since it is way obvious the statement since it does not add any information: “ As is well known, the premise of plant somatic embryogenesis is that embryogenic callus is induced and can proliferate stably; thus, it is necessary to induce stably proliferating embryogenic callus in hybrid larch.

Answer: “ As is well known, the premise of plant somatic embryogenesis is that embryogenic callus is induced and can proliferate stably; thus, it is necessary to induce stably proliferating embryogenic callus in hybrid larch“ has been deleted.

(13) Line 330: After the co-culture, numerous Agrobacterium were symbiotic on the surface of embryogenic callus. Symbiotic, REALLY?

Answer: ”symbiotic” has been replaced with “appeared”.

(14) Line 858-859: What does ‘ E: cotyledons germinate’ mean?

Answer: ”cotyledons germinate” has been replaced with “somatic embryo germination”.

(15) Figures 1, 2, 4, 8 and 9 do lack size reference bars.

Answer: ”Figures 1, 2, 4, 8 and 9” has been added the size reference bars.

(16) The Discussion sounds more like a Literature Review.... (see lines 357 to 372). It is poorly elaborated as presented.

Answer: The ”Discussion” section has been revised in this manscript.

(17) Lines 358 and 362: Stasolla summarized ... Stasolla believe... Authors shouls sitck to a scientific writing style. Even though it should appear Satasolla ang Yeung [ ]

Answer: They have been revised in this manscript.

(18) Line 372: … stage was the progeny stage [32]. What does progeny stage mean?

Answer: ”the progeny stage” was incorrect, it has been deleted and revised in this manscript. 

(19) Lines 372-389: In what sense authors want to stress the collection dates/times? Does it really matter? In my opinion the stage is what really matters. Above all, authors do not provide a convincing explanation to justify this difference. The discussion is way superficial.

Answer: The collection date actually corresponds to the development stage, and this difference actually was presented in Table 5. It also has been revised in this manscript.

(20) Line 386: This finding is also consistent with the results reported in previous studies. Please, specify them!

Answer: They have been specified in this manscript.

(21) And this also applies to the dose-response data (lines 392-423) during induction step. The Discussion again sounds as a literature review. The finding are not really compared and contrasted with existing and referenced literature.

Answer: ” the dose-response data” and ”Discussion” have been revised in this manscript.

(22) If one have a look from line 441-454 this very same style appears denoting a literature review on the topic (Agrobacterium-mediated transformation).

Answer: They have been revised in this manscript.

(23) For transformation purposes why authors did not consider adding acetosyringone or coniferyl alcohol as vir gene inducers? How about the Agro strain GV 3101 used as compared to others - C58 /EHA 105? It would be interesting to consider for the Discussion.

Answer: We did not consider adding acetosyringone or coniferyl alcohol as vir gene inducers because of refering to the study on Larix olgensis and the reference [54] has been added in this manscript. In addition, GV3101 and other stains has been added in discussion section. 

Yours Sincerely 

Sufang Zhang

---

## [Editor Report · Decision Letter 1]

4 Oct 2021

Embryogenic callus induction from immature zygotic embryos and genetic transformation of Larix kaempferi 3x Larix gmelinii 9

PONE-D-21-16841R1

Dear Dr. Zhang,

We’re pleased to inform you that your manuscript has been judged scientifically suitable for publication and will be formally accepted for publication once it meets all outstanding technical requirements.

Kind regards,

Vijay Kumar

Academic Editor

PLOS ONE

Additional Editor Comments (optional):

authors has revised the manuscript significantly and now I recommend it for acceptance in the current state.
---

## [Editor Report · Acceptance letter]

7 Oct 2021

PONE-D-21-16841R1 

Embryogenic callus induction from immature zygotic embryos and genetic transformation of *Larix kaempferi* 3x *Larix gmelinii* 9 

Dear Dr. Zhang:

I'm pleased to inform you that your manuscript has been deemed suitable for publication in PLOS ONE. Congratulations! Your manuscript is now with our production department. 

Kind regards, 

on behalf of

Dr. Vijay Kumar 

Academic Editor

PLOS ONE